# Stagewise Development in Transformers and the Geometry of the Loss Landscape

## Abstract

Deep learning involves navigating a high-dimensional parameter space guided by the loss landscape. In the process, complex computational structures form and re-form inside the neural network, leading to shifts in input–output behavior. It is a priority for the science of deep learning to uncover principles governing the development of neural network structure and behavior. Drawing from the framework of singular learning theory, we propose that model development is governed by the local geometry of the loss landscape. We investigate this link by monitoring the geometry of the loss landscape throughout training for transformers trained as language models or for a synthetic in-context regression task. We divide training into "developmental stages" marking discrete shifts in loss landscape geometry. We then confirm that these stages coincide with significant changes in the internal computational structure and the input–output behavior of our models. Our findings provide new insights into transformer development and underscore the potential of a geometric perspective for understanding modern deep learning.

## 1 Introduction

In modern deep learning, a striking phenomenon is the event of sudden shifts in a model's internal computational structure and associated changes in generalization behavior (e.g., Wei et al., 2022; Olsson et al., 2022; McGrath et al., 2022). As large models become more deeply integrated into real-world applications, understanding this phenomenon is a priority for the science of deep learning.

While the scale and importance of this phenomenon is new, the idea that neural network learning proceeds in discrete developmental stages goes back decades (Raijmakers et al., 1996). In the case of deep linear networks, we know that training proceeds through stages in which the model learns progressively higher-rank approximations of the data distribution (Baldi and Hornik, 1989; Rogers and McClelland, 2004; Saxe et al., 2019). Unfortunately, it is not clear how this perspective on stagewise development can be generalized to large nonlinear models such as transformers which exhibit more complex internal computational structure and are trained on more complex datasets.

We propose that the key to understanding stagewise development is the geometry of the loss landscape. This perspective on neural network development is motivated by Singular Learning Theory (SLT; Watanabe, 2009). SLT proves that, for Bayesian neural networks, the posterior develops in discrete developmental stages in a so-called *singular learning process* governed by the local geometry of the model likelihood (Watanabe, 2009; Chen et al., 2023). While there are many differences between Bayesian deep learning and modern neural network training, we believe that the link between local geometry and stagewise development is fundamental to learning.

In this paper, we contribute a thorough empirical investigation into the relationship between loss landscape geometry and stagewise development. In particular, we train transformers in two distinct learning settings, and we demonstrate changes in their loss landscape geometry can be associated with changes in their internal computational structure and input–output behavior. This finding provides evidence that loss landscape geometry is closely linked to neural network development in modern deep learning.

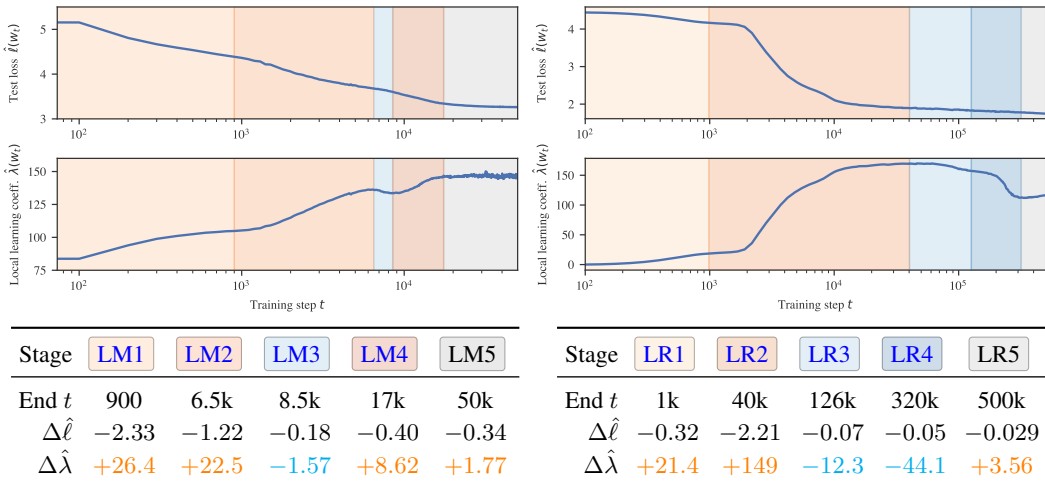

|       | Stage | LM1 | LM2 | LM3 | LM4 | LM5 |
|-------|-------|-----|-----|-----|-----|-----|
| End $t$ |     | 900 | 6.5k | 8.5k | 17k | 50k |
| $\Delta\hat{\ell}$ |  | $-2.33$ | $-1.22$ | $-0.18$ | $-0.40$ | $-0.34$ |
| $\Delta\hat{\lambda}$ |  | $+26.4$ | $+22.5$ | $-1.57$ | $+8.62$ | $+1.77$ |

|       | Stage | LR1 | LR2 | LR3 | LR4 | LR5 |
|-------|-------|-----|-----|-----|-----|-----|
| End $t$ |     | 1k  | 40k | 126k | 320k | 500k |
| $\Delta\hat{\ell}$ |  | $-0.32$ | $-2.21$ | $-0.07$ | $-0.05$ | $-0.029$ |
| $\Delta\hat{\lambda}$ |  | $+21.4$ | $+149$ | $-12.3$ | $-44.1$ | $+3.56$ |

(a) Two-layer attention-only language transformer.   (b) In-context linear regression transformer.

Figure 1: We train transformer models on both (a) natural language data and (b) synthetic in-context linear regression data. In addition to test loss (top row), we track loss landscape geometry as quantified by the Local Learning Coefficient (LLC) (middle row, see Section 3). Critical points in the LLC curve mark boundaries between distinct *developmental stages* (ranges of training time) (bottom row, indicated with orange hues for increasing LLC or blue for decreasing LLC, see Section 4). We show in Sections 5 and 6 that these stages coincide with the formation of certain internal structures or changes in input–output behavior. These stage divisions are specific to these training runs, but we show in Appendix B.1 that similar divisions arise when training with different seeds.

In more detail, we propose the following geometry-based stage identification methodology:

1. **Transformer training** (Section 2): We train transformers in two learning settings, namely *language modeling*, a transformer with around 3M parameters modeling a subset of the Pile (Gao et al., 2020; Xie et al., 2023), and *in-context linear regression*, a transformer with around 50k parameters modeling synthetic regression data following Garg et al. (2022).

2. **Local geometry tracking** (Section 3): We quantify the local geometry of the loss landscape at frequent checkpoints by estimating the Local Learning Coefficient (LLC; Lau et al., 2024), a measure of local geometric complexity derived from SLT (Watanabe, 2009).

3. **Geometry-based stage division** (Section 4): Motivated by the singular learning process in Bayesian inference (Watanabe, 2009; Chen et al., 2023), we search for critical points in the LLC curve across training time and use these as boundaries to divide training into stages.

4. **Stage validation** (Sections 5 and 6): To show that these divisions are meaningful, we track shifts in each model's internal computational structure and input–output behavior across checkpoints, quantified using various setting-specific metrics.

Following this methodology reveals that each transformer's training can be clearly divided into several developmental stages (see Figure 1). Moreover, we find that these divisions are *meaningful* in that the stages coincide with significant, interpretable shifts in the internal computational structure and input–output behavior of the models (Sections 5 and 6).

These results have several implications, discussed further in Section 8. Both our methodology for stage identification as well as the specific stages we identify for these transformers contribute to the growing literature on understanding transformer behavior, structure, and training dynamics. Furthermore, we find several stages in which the loss landscape geometry becomes *increasingly degenerate*, indicating the model becoming simpler—a phenomenon not predicted by prior models of saddle-to-saddle dynamics. Finally, the fact that simply monitoring the local geometry of the loss landscape reveals meaningful developmental stages for modern transformer models in two distinct learning settings points to a fundamental link between local geometry and development, and the crucial role geometry has the potential to play in advancing our understanding of deep learning.

## 2 TRAINING TRANSFORMERS IN TWO SETTINGS

We study transformers trained in two different learning settings, namely *language modeling* and *in-context linear regression*. These settings have been the subject of recent work on the emergence of In-Context Learning (ICL), a compelling example of a sudden shift in a model's internal computational structure in modern deep learning (Olsson et al., 2022).

In this section, we describe both settings and introduce their data distributions and loss functions. Common to both settings is a transformer model denoted $f_w$ with parameters $w$, which takes as input a sequence of tokens, also called a context. We describe specific architecture details and training hyperparameters in Appendices D.1 and D.2.

**Language modeling.** Elhage et al. (2021) and Olsson et al. (2022) observed that two-layer attention-only transformers (transformers without MLP layers) form interesting internal computational structures supporting ICL, including induction heads. In order to compare with the behavioral and structural analysis of these prior works we adopt the same architecture.[1]

We consider the standard task of next-token prediction for sequences of tokens taken from a subset of the Pile (Gao et al., 2020; Xie et al., 2023). We denote the input context, a sequence of tokens $t_k$, by $S_K = (t_1, \ldots, t_K)$ where $K$ is the context length. We denote by $S_{\leq k}$ the sub-sequence $(t_1, \ldots, t_k)$ of the context $S_K$. Our data is a collection of length-$K$ contexts, $\{S_K^i\}_{i=1}^n$; the superscript $i$ indicates the $i$-th sample in a total of $n$ such samples. For $1 \leq i \leq n$, the notation $S_{\leq k}^i$ should be understood as the sub-sequence of the $i$-th context, $S_K^i$.

Given a context $S_{\leq k}^i$, the transformer model $f_w$ outputs a vector of logits $f_w(S_{\leq k}^i)$ such that $\text{softmax}(f_w(S_{\leq k}^i))$ is a probability distribution over all tokens (we denote by $\text{softmax}(f_w(S_{\leq k}^i))[t]$ the probability of token $t$). The *per-token empirical loss* for language modeling is then the average cross-entropy between this distribution and the true next token at each index $k \in \{1, \ldots, K-1\}$,

$$\ell_{n,k}(w) = -\frac{1}{n} \sum_{i=1}^n \log\left(\text{softmax}(f_w(S_{\leq k}^i))[t_{k+1}^i]\right). \tag{1}$$

The associated *empirical loss* is $\ell_n(w) = \frac{1}{K-1} \sum_{k=1}^{K-1} \ell_{n,k}(w)$, with the *test loss* $\hat{\ell}$ defined analogously on a held-out set of examples. The corresponding *population loss* $\ell(w)$ is defined by taking the expectation with respect to the true distribution of contexts (see also Appendix A.4).

**In-context linear regression.** Following the framework of Garg et al. (2022), a number of recent works have explored ICL in the setting of learning simple function classes, such as linear functions. This setting is of interest because we understand theoretically optimal ICL, and because simple transformers are capable of good ICL performance in practice.

We give a standard presentation of transformers trained for in-context linear regression. A *task* is a vector $\mathbf{t} \in \mathbb{R}^D$. Given a task $\mathbf{t}$, we generate $x_i \in \mathbb{R}^D, y_i \in \mathbb{R}$ iid for $i = 1, \ldots, K$ according to the joint distribution $q(x, y|\mathbf{t}) = q(y|x, \mathbf{t})q(x)$, resulting in the context $S_K = (x_1, y_1, \ldots, x_{K-1}, y_{K-1}, x_K)$ with label $y_K$. Note that the $x_i$'s and $y_i$'s are yet to be tokenized in this notation (see Appendix D.2.2). A sub-sequence of $S_K$ is denoted $S_{\leq k}$ as above with $S_{\leq k} = (x_1, y_1, \ldots, x_k)$ with label $y_k$. We study the setting with task distribution $q(\mathbf{t}) = \mathcal{N}(0, I_D)$, input distribution $q(x) = \mathcal{N}(0, I_D)$, and output distribution $q(y|x, \mathbf{t}) = \mathcal{N}(\mathbf{t}^T x, \sigma^2)$.

Consider a set of samples $\{(\mathbf{t}_i, S_K^i, y_K^i)\}_{i=1}^n$ which consists of $n$ iid samples drawn as described above. Upon running a context through the transformer, we obtain a prediction $\hat{y}_k^i = f_w(S_{\leq k}^i)$ for each sub-sequence $S_{\leq k}^i$, which leads to the *per-token empirical loss* for in-context linear regression,

$$\ell_{n,k}(w) = \frac{1}{n} \sum_{i=1}^n (\hat{y}_k^i - y_k^i)^2, \tag{2}$$

for $1 \leq k \leq K$. The associated empirical loss is $\ell_n(w) = \frac{1}{K} \sum_{k=1}^K \ell_{n,k}(w)$. The in-context linear regression test loss and population loss are defined analogously.

---

[1]We also consider one-layer attention-only transformers in Appendix B.5. We don't study transformers with MLP layers in this setting, though we do use MLP layers in the in-context linear regression setting.

## 3 QUANTIFYING GEOMETRY VIA THE LOCAL LEARNING COEFFICIENT

We track the evolution of degeneracy in the local geometry of the loss landscape throughout training by estimating the Local Learning Coefficient (LLC; Watanabe, 2009; Lau et al., 2024) at model checkpoints. In this section we review the LLC and the estimation procedure of Lau et al. (2024).

**The Local Learning Coefficient (LLC).** Given a local minimum $w^*$ of the population loss $\ell$, the LLC of $w^*$, denoted $\lambda(w^*)$, is a positive rational number measuring the amount of *degeneracy* in the geometry of $\ell$ near $w^*$ (Watanabe, 2009; Lau et al., 2024). Intuitively, the geometry is more degenerate (lower LLC) at a parameter $w^*$ if there are more ways in which $w$ can be varied near $w^*$ such that $\ell(w)$ remains equal to $\ell(w^*)$. More formally, the LLC of a local minimum $w^*$ of the population loss $\ell$ (a negative log likelihood) can be defined based on the asymptotics of the volume $V(\epsilon)$ of the set of nearby parameters with loss less than $\ell(w^*) + \epsilon$. As $\epsilon \to 0$ this volume scales asymptotically in $\Theta\left(\epsilon^{\lambda(w^*)} \log(1/\epsilon)^{m(w^*)-1}\right)$ where $\lambda(w^*)$ is the LLC and $m(w^*)$ is another geometric quantity called the *local multiplicity*. We refer readers to Figure 2 for a conceptual illustration, Appendix A.1 for additional discussion, and Lau et al. (2024) for full formal treatment.

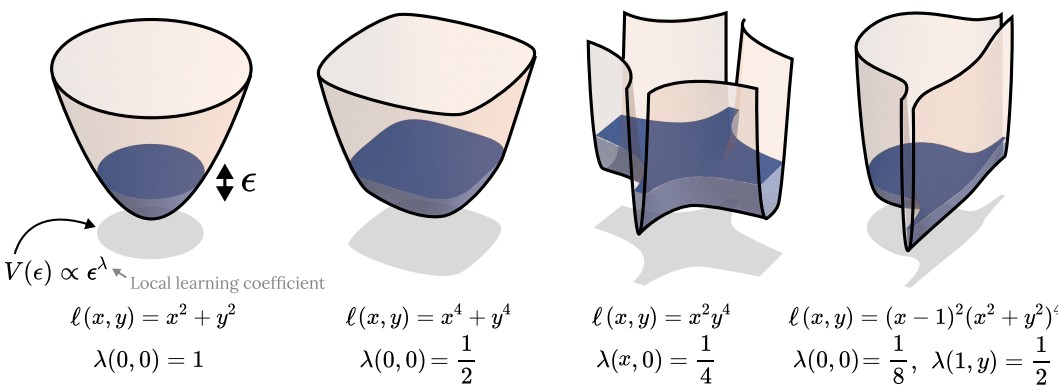

$$V(\epsilon) \propto \epsilon^\lambda \quad \text{Local learning coefficient}$$

| $\ell(x,y) = x^2 + y^2$ | $\ell(x,y) = x^4 + y^4$ | $\ell(x,y) = x^2 y^4$ | $\ell(x,y) = (x-1)^2(x^2+y^2)^4$ |
| --- | --- | --- | --- |
| $\lambda(0,0) = 1$ | $\lambda(0,0) = \dfrac{1}{2}$ | $\lambda(x,0) = \dfrac{1}{4}$ | $\lambda(0,0) = \dfrac{1}{8},\ \lambda(1,y) = \dfrac{1}{2}$ |

Figure 2: Increasingly degenerate population loss landscapes for a two-dimensional parameter space. The Local Learning Coefficient (LLC) can be defined as an asymptotic volume scaling exponent—the order at which the parameter space volume, within a given neighborhood and with a given maximum loss, shrinks as the maximum loss threshold is reduced to that of the local minimum (in these examples, the local multiplicity is 1). See Appendix A.1 for more details.

**Estimating the LLC.** Lau et al. (2024) introduced an estimator for the LLC based on stochastic-gradient Langevin dynamics (SGLD; Welling and Teh, 2011), which we use in our experiments. Let $w^*$ be a local minimum of the population loss $\ell$. The LLC estimate $\hat{\lambda}(w^*)$ is

$$\hat{\lambda}(w^*) = n\beta \left[ \mathbb{E}^\beta_{w|w^*,\gamma}[\ell_n(w)] - \ell_n(w^*) \right], \tag{3}$$

where $\mathbb{E}^\beta_{w|w^*,\gamma}$ denotes the expectation with respect to the Gibbs posterior

$$p(w; w^*, \beta, \gamma) \propto \exp\left\{ -n\beta\ell_n(w) - \frac{\gamma}{2}||w - w^*||_2^2 \right\}$$

with inverse temperature $\beta$ (controlling the contribution of the empirical loss landscape) and localization strength $\gamma$ (controlling proximity to $w^*$). Intuitively, the more degenerate the geometry, the more ways there are to vary $w$ near $w^*$ without changing the loss, the easier it is for a sampler exploring the Gibbs posterior to find points of low loss, and, in turn, the lower $\hat{\lambda}(w^*)$. Appendix A.2 discusses technical SGLD details, Appendix D.3 outlines our hyperparameter tuning procedure, and Appendices D.1.4 and D.2.4 document the hyperparameters used in our experiments.

**Limitations in LLC estimation.** Strictly speaking, the LLC is defined only at local minima for loss functions arising as negative log likelihoods. By using the estimator at arbitrary transformer training checkpoints with a loss function based on overlapping contexts, we stray beyond these assumptions. Nevertheless, consistent with the findings of Chen et al. (2023), the estimator appears to produce reliable results throughout training in practice (see Appendices A.3 and A.4 for further discussion).

## 4 THE SINGULAR LEARNING PROCESS AND STAGEWISE DEVELOPMENT

We use critical points (that is, plateaus, where the first derivative vanishes) in the LLC curve to define *stage boundaries* that divide training into *developmental stages*. This approach to stage identification is motivated by the singular learning process in Bayesian inference. In this section we review the singular learning process and outline our stage identification approach in more detail.

**The singular learning process.**   Watanabe's free energy formula (Watanabe, 2018, Theorem 11), generalized to a local setting by Chen et al. (2023), gives an asymptotic expansion in the number of samples $n$ of the *Bayesian free energy* $F_n$ of some neighborhood $W^*$ surrounding a local minimum of $\ell$, $w^* \in W^*$, in terms of the empirical loss $\ell_n$, the LLC $\lambda$, and the local multiplicity $m$:

$$F_n(W^*) = n\ell_n(w^*) + \lambda(w^*)\log n - (m(w^*) - 1)\log\log n + O_p(1). \quad (4)$$

The free energy of a neighborhood is related to the integral of the Bayesian posterior on the neighborhood by a negative logarithm, so the lower the free energy the more likely the neighborhood is according to Bayes' rule (the prior contributes only in the sub-leading terms as long as it is positive).

The coefficients of the linear and logarithmic terms are the empirical loss (a negative log likelihood) and the LLC, respectively. This creates a trade-off between accuracy ($\ell_n$) and degeneracy ($\lambda$) that changes as $n$ increases. At certain *critical dataset sizes* the neighborhood having the lowest free energy may rapidly change, causing the Bayesian posterior to suddenly "jump" from concentrating around one local minima to another (illustrated in Figure 3). The sequence of discrete posterior jumps in Bayesian inference is referred to as the *singular learning process* (Watanabe, 2009, §7.6).

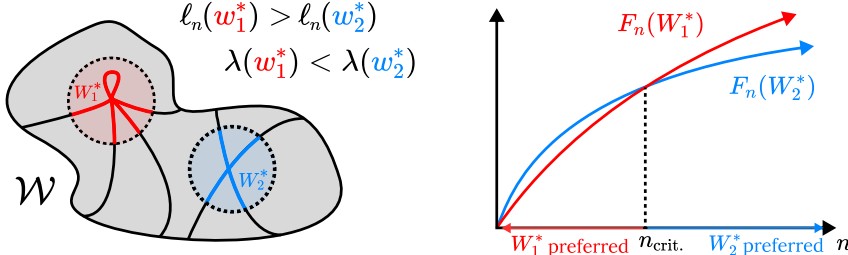

Figure 3: Conceptual illustration of a discrete "jump" in the Bayesian posterior as the number of samples increases. Watanabe's free energy formula sets up a trade-off between the linear term (with coefficient $\ell_n$, the loss) and the logarithmic term (with coefficient $\lambda$, the LLC). Consider two local minima $w_1^*, w_2^*$ with neighborhoods $W_1^*, W_2^*$. As the number of samples $n$ increases, if $w_2^*$ has lower loss and higher LLC than $w_1^*$, its neighborhood $W_2^*$ may suddenly achieve lower free energy than $W_1^*$, causing the Bayesian posterior to rapidly shift from to concentrating in $W_1^*$ to $W_2^*$.

**LLC plateaus separate developmental stages.**   The connection between the singular learning process in Bayesian inference and stagewise development in deep learning is not understood in general, but has been studied in small autoencoders by Chen et al. (2023). Chen et al. (2023) showed that in these models both Bayesian inference and stochastic gradient descent undergo sudden transitions between various encoding schemes, reflected as sudden changes in the estimated LLC, as predicted by Watanabe's free energy formula.

This perspective suggests that changes in the local geometry of the population loss, as measured by the LLC, reflect qualitative changes in the network parameter. However, in larger models, we expect these changes to be more gradual. Accordingly, instead of phase transitions, we speak of *developmental stages* separated by *stage boundaries* at which the posterior is stably concentrated around a given local minima. What remains is that plateaus in the estimated LLC curve indicate boundaries between distinct qualitative changes in the network parameter. This motivates our approach of using such plateaus to divide training into developmental stages.

In our experiments, we identify plateaus in the estimated LLC over checkpoints as follows. We first lightly smooth the LLC curve with a Gaussian process (to facilitate stable numerical differentiation). We then numerically differentiate the smoothed curve with respect to log training time. We identify plateaus as approximate zeros of this derivative, namely local minima of the absolute value of the derivative that fall below a small threshold. Appendix A.5 discusses this procedure in more detail.

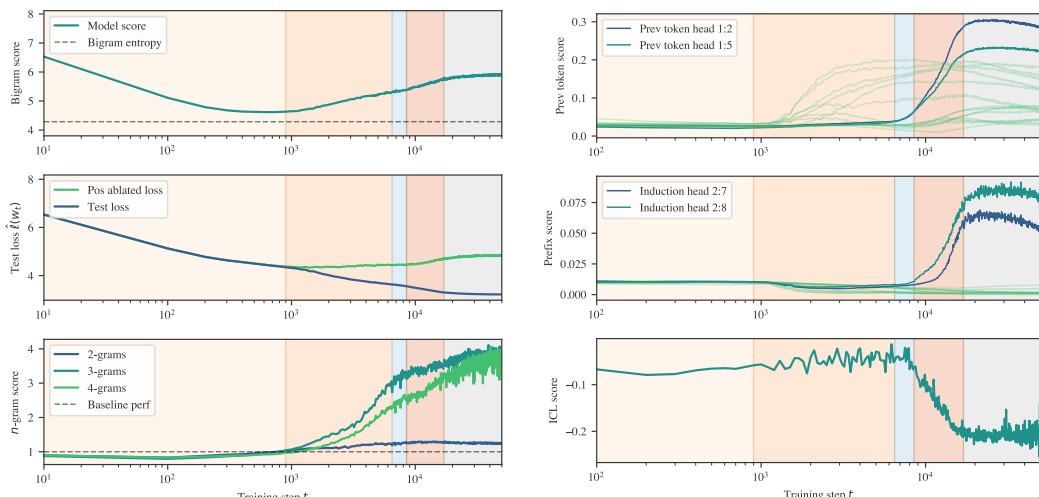

(a) The language transformer learns bigram statistics in LM1 (*top*). At the start of LM2, the positional embedding suddenly becomes useful (*middle*), enabling behavioral changes such as the learning of common *n*-grams (*bottom*).

(b) Induction circuit formation begins with previous-token heads (LM3, *top*), followed by induction heads (LM4, *middle*), leading to a drop in ICL score (LM4, *bottom*). The *h*th attention head in layer $l$ is indexed as $l : h$. *Top and middle*: green lines indicate scores for all other heads.

Figure 4: Stagewise development in small language models.

## 5 RESULTS FOR LANGUAGE MODELING

Plateaus in LLC estimates (Figure 1a) reveal five developmental stages over the training of our language transformers. We associate these stages with the development of bigrams (LM1), *n*-grams (LM2), previous-token heads (LM3), and induction heads (LM4). There may be other interesting developmental changes; we do not claim this list is exhaustive. We did not, for example, discover significant changes in stage LM5.

It was previously known that small attention-only transformer language models learn bigrams (0-layer models), skip-trigrams (1-layer models), and induction circuits (2-layer models) (Olsson et al., 2022). This represents a progression of internal structure in the final trained transformer, *as a function of increasing depth*. An original contribution of the present paper is to clearly identify the following progression of developmental stages *over training* in this small language model:

**Bigrams → $n$-grams → Previous-token heads → Induction heads**.

In this section, we present the details for a particular seed, see Appendix B.1 for the other seeds.

### 5.1 STAGE LM1 (0–900 STEPS)

**Behavioral changes.** The model learns bigram statistics, which is optimal for single-token prediction. Figure 4a (top) shows that the average cross entropy between logits and empirical bigram frequencies (see Appendix B.3.1) is minimized at the LM1–LM2 boundary, with a value only .3 nats above the entropy of the empirical bigram distribution.

### 5.2 STAGE LM2 (900–6.5K STEPS)

**Behavioral changes.** A natural next step after bigrams are $n$-*grams*, token sequences of length $n$. We define an $n$-*gram score* as the ratio of final-position token loss on (1) a baseline set of samples from a validation set truncated to $n$ tokens and (2) a fixed set of common $n$-grams (see Appendix B.3.2). We see a large improvement in $n$-gram score for $n = 3, 4$ in Figure 4a (bottom), rising to several times the baseline ratio (1.0). Although this is one natural next step for the learning process, we do not rule out other possible developmental changes for this stage, such as skip-trigrams.

(a) LM1 (0 - 900)

<lendoftextl>I should like, before proceeding further, to tell you how I feel about the State which we have described. I might compare myself to a person who, on beholding beautiful animals either created by the painter's art, or, better still, alive but at rest, is seized with a desire of seeing them in motion or engaged in some struggle or conflict to which their forms appear suited;

(b) LM2 (900 - 6,500)

<lendoftextl>In the midst of unexpected circumstances with Linux and Python, the honorable Supreme Court in Boston delivered a ruling emphasizing a crazy database framework last week.

(c) LM3 + LM4 (6,500 - 17,000)

<lendoftextl>Mr. and Mrs. Dursley, of number four, Privet Drive, were proud to say that they were perfectly normal, thank you very much. They were the last people you'd expect to be involved in anything strange or mysterious, because they just didn't hold with such nonsense. Mr. Dursley was the director of a firm called Grunnings, which made drills. He was a big, beefy man with hardly any neck, although he did have a very large mustache. Mrs. Dursley was thin and blonde and had nearly twice the usual amount of neck, which came in very useful as she spent so much of her time craning over garden fences, spying on the neighbors. The Dursleys had a small son called Dudley and in their opinion there was no finer boy anywhere.

Figure 5: Samples are shown with tokens highlighted to indicate changes in logits during a given range. Red is improved performance (higher logit output for the true next token) and blue is worse. Sample (a): improvement in bigrams (LM1) such as "te/ll, ab/out, des/ire, mot/ion, eng/aged, strugg/le, etc." Sample (b): improvement in common $n$-grams (LM2) such as "L/in/ux, P/y/th/on, h/on/or/able, S/up/reme, dat/ab/ase, f/ram/ew/ork." Sample (c): development of in-context learning via induction circuits (LM3, LM4), visible in the improved predictions in the word "D/urs/ley" after the first time it appears in the context, as initially observed by (Olsson et al., 2022).

**Structural changes.** The positional embedding is necessary for learning $n$-grams, and, as expected, the model becomes dependent on the positional embedding during LM2. This is apparent in comparing the test loss with and without the positional embedding zero-ablated in Figure 4a (middle)—the curves are indistinguishable at first but diverge at the LM1–LM2 boundary (see Appendix B.4.1). We also see a rise in previous-token attention among second layer attention heads in the background of Figure 4b (top), which we also suspect plays a role with $n$-grams.

Interestingly, even before the heads that eventually become induction heads develop their characteristic attention patterns in stages LM3 and LM4, they begin to compose (that is, read and write from the same residual stream subspace) near the start of stage LM2 (see Figure B.5 and Appendix B.4.2). This suggests that the foundations of the induction circuit model are laid well in advance of any measurable change in model outputs or attention activations.

## 5.3 STAGE LM3 (6.5K–8.5K STEPS)

**Structural changes.** The first half of the induction circuit (Elhage et al., 2021) begins to form in this stage. Figure 4b (top) shows that for the previous-token heads that will later participate in the circuit (highlighted in blue), the fraction these heads attend to the immediately preceding token begins to increase during this stage (see Appendix B.4.3). During this stage the LLC decreases, suggesting an increase in degeneracy and decrease in model complexity, perhaps related to the interaction between heads. It would be interesting to study this further via mechanistic interpretability.

## 5.4 STAGE LM4 (8.5K–17K STEPS)

**Behavioral changes.** The model learns to perform ICL as studied by Olsson et al. (2022) (Figure 4b bottom).

**Structural changes.** The second half of the induction circuits, second-layer induction heads, begin to develop. Given a sequence $[A][B] \ldots [A]$, the *prefix-matching score* of Olsson et al. (2022) measures attention to $[B]$ from the latter $[A]$ (see Appendix B.4.4). Figure 4b (middle) shows that the prefix-matching score increases for the two heads that become induction heads (highlighted in blue).

## 5.5 VISUALIZING BEHAVIORAL CHANGES

In Figure 5, we visualize changes in the model's input-output behavior by comparing model predictions before and after developmental stages and highlighting tokens with the greatest differences.

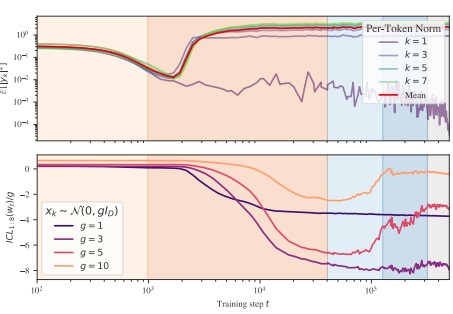 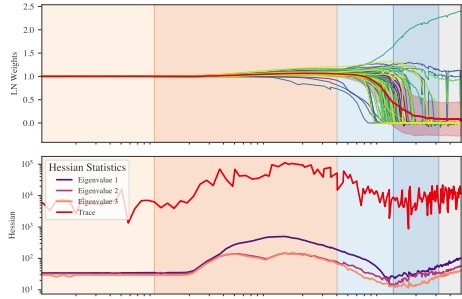

(a) *Top*: During LR1, the model learns to predict with the task prior, $x_k \mapsto \hat{y}_k = 0$. *Bottom*: ICL emerges during LR2. In LR3, the model becomes worse at ICL on OOD inputs $x_k \sim \mathcal{N}(0, gI_D)$.

(b) *Top*: During LR3 and LR4, layer normalization weights "collapse," possibly contributing to an LLC decrease. *Bottom*: Hessian-based statistics reveal only one stage boundary (Appendix C.2.2).

Figure 6: Stagewise development in linear regression transformers.

## 6 RESULTS FOR LINEAR REGRESSION

In the linear regression setting, plateaus in the LLC estimate (Figure 1b) reveal five developmental stages, corresponding to learning the task prior (LR1), acquiring in-context learning (LR2), and "over-fitting" to the input distribution (LR3/LR4). We present the stage analysis for one seed, see Appendix B.1 for the other seeds.

### 6.1 STAGE LR1 (0–1K STEPS)

**Behavioral changes.** Similar to bigrams in the language model setting, the model learns the optimal context-independent algorithm, which is to predict using the average task $\bar{\mathbf{t}}$, which is zero for our regression setup. Figure 6a shows that the average square prediction for all tokens $\mathbb{E}[\|\hat{y}_k\|^2]$ decreases during LR1, reaching a minimum of 0.1 (smaller than the target noise $\sigma^2 = 0.125$) slightly after the end of LR1.

### 6.2 STAGE LR2 (1K–40K STEPS)

**Behavioral changes.** The model acquires in-context learning during this stage (Figure 6a bottom, with input gain $g = 1$). This parallels stage LM4 in the the language model setting.

**Structural changes.** The token and positional embedding begin to "collapse" towards the end of this stage, effectively losing singular values and aligning with the same activation subspace (Appendix C.4.1). At the same time, several attention heads form distinct and consistent patterns (Appendix C.4.5).

### 6.3 STAGES LR3 & LR4 (40K–126K & 126K–320K STEPS)

**Behavioral changes.** The model begins to "overfit" to the input distribution: performance continues to improve on typical samples, but deteriorates on extreme samples for which the norm of the inputs $x_k$ is larger than encountered in during training (Figure 6a with input gain $g \neq 1$).

**Structural changes.** A large fraction of the layer normalization weights rapidly go to zero in this stage (Figure 6b and Figure C.6). We term this *layer normalization collapse*. The phenomenon is most pronounced in the unembedding layer normalization, where it occurs in tandem with a similar collapse in the weights of the unembedding matrix. This results in the model learning to read its prediction $y$ from a handful of privileged dimensions of the residual stream. Since this means that the network outputs become insensitive to changes in many of the parameters, we conjecture that this explains part of the striking decrease in estimated LLC over these stages (see Appendix C.4.3). Stage LR4 differs from LR3 in the layer normalization collapse expanding from the unembedding to earlier layer norms, particularly the layer normalization before the first attention block. This affects a smaller fraction of weights than the unembedding collapse (Appendix C.4.4).

## 7 RELATED WORK

**Loss landscape geometry in deep learning.** Given the central role played by the loss landscape in deep learning it is unsurprising that there have been many attempts to study its geometry. One approach is to visualize low-dimensional slices of the loss landscape (Erhan et al., 2010; Goodfellow et al., 2014; Lipton, 2016; Li et al., 2018; Notsawo et al., 2024). Unfortunately, a random slice is with high probability a quadratic form associated to nonzero eigenvalues of the Hessian and is thus biased against geometric features that we know are important, such as degeneracy. Antognini and Sohl-Dickstein (2018) have emphasized the difficulty of probing the loss landscape of neural networks with dimensionality reduction tools.

Standard methods of quantifying the geometry of the loss landscape, such as by studying the Hessian itself, also fail to quantify important aspects of degeneracy. For example, the Hessian trace or maximum eigenvalues quantify the curvature of a critical point but ignore its degenerate dimensions. In Figure 6b and Appendix C.2.2 we show that these metrics are unable to detect most of the stage boundaries detected by the LLC in our in-context linear regression setting. The Hessian rank accounts for degenerate directions but fails to distinguish between directions by the order of their degeneracy. In contrast, the LLC is a quantitative measure of loss landscape geometry that directly accounts for different kinds of degeneracy (see Appendix A.1).

**Stagewise development and geometry in nonlinear dynamics and developmental biology.** Neural network training is a stochastic dynamical system, in which the governing potential (the population loss) encodes the structure of the data distribution along with the constraints of the network architecture. It is well-understood in nonlinear dynamics that the local geometry of a potential can give rise to stagewise development of structure in the system (Waddington, 1957; Thom, 1988, see also Franceschelli, 2010). This connection between geometry and stagewise development has been observed in biological systems at significant scale and in the presence of stochasticity (Freedman et al., 2023). We have emphasized changes in geometry *over a stage* whereas in developmental biology the focus, in the mathematical framework of bifurcation theory, is more on the singular geometry *at stage boundaries* (Rand et al., 2021; MacArthur, 2022; Sáez et al., 2022).

**"Developmental stages" versus "phase transitions."** Many works on emergent phenomena use the term "phase transition" to label a rapid change in model structure or behavior. Our developmental stages can also be modeled as phase transitions (between phases inhabited by the model at stage boundaries). We adopt the terminology from biology to avoid confusion, since in some cases our developmental stages occur over a large number of training steps, whereas the term "phase transition" usually connotes a rapid change. However, we note that phase transitions *need not* be rapid in time: from a mathematical point of view (Gilmore, 1981; Chen et al., 2023), a phase transition is a shift in the configuration of a system (or a distribution over such configurations) from the neighborhood of one critical point of a potential to that of another, occurring rapidly as a function of *some control variable*. The control variable may be time, but it need not be. In developmental biology, there are carefully modeled phase transitions that take place over *days* in real time, but rapidly with respect to an inferred *developmental time* (Freedman et al., 2023).

## 8 DISCUSSION

In this paper, we have contributed a detailed examination of the development of transformer models in two distinct learning settings. We quantified the changing degeneracy of the geometry of the population loss throughout transformer training by estimating the local learning coefficient (LLC). Based on an analogy introduced by Chen et al. (2023) between deep learning and the singular learning process, we divided these training runs into developmental stages at critical points of the LLC curve. We also monitored the internal computational structure and input–output behavior of our transformers throughout training, as quantified through a range of setting-specific metrics. We found an approximate correspondence between developmental stages identified based on changes in loss landscape degeneracy and significant structural and behavioral changes taking place in each model. In this section, we discuss several implications of these findings.

**Towards a geometric understanding of deep learning.** Our main finding is an approximate correspondence between developmental stages identified via loss landscape geometry and significant structural and behavioral changes identified via setting-specific analyses. This correspondence is evidence that the development of our transformers is closely linked to the geometry of the loss landscape, underscoring the potential of loss landscape geometry, and particularly its degeneracy as quantified by the LLC, as a crucial lens through which to study the development of deep learning models. While we have illustrated this correspondence in two distinct learning settings, including language modeling with a nontrivial transformer architecture, it remains necessary for future work to verify this link for larger-scale models across a more diverse range of emergent phenomena.

**Developmental interpretability.** We have shown that a geometry-based stage identification methodology can reveal meaningful changes in our transformers. We note that our analysis is not exhaustive. Indeed, we expect that in general only certain "macroscopic" changes, such as the emergence of in-context learning, will be delineated by plateaus in the estimated LLC. Building on our approach, Wang et al. (2024) have shown promising results by measuring the LLC with respect to certain network components and with different data distributions, providing a refined picture of model development. We are optimistic that further work in this direction will lead to geometry-based tools that can offer insights into the development of larger models.

Monitoring model development through loss landscape geometry offers an alternative to monitoring *progress measures* such as those derived by Barak et al. (2022) or developed using mechanistic insights from similar models by Nanda et al. (2023). Both approaches can reveal developments not visible in the loss curve. Monitoring loss landscape geometry is setting-agnostic and able to detect developments before having mechanistically understood the end result. Of course, once a development is detected through a change in geometry, it remains to interpret what has changed.

**Cases studies in transformer development.** The specific developments we observe in each setting contribute to the growing empirical literature on the emergence of in-context learning in transformers. We replicate the emergence of induction heads in two-layer attention-only transformer language models (Elhage et al., 2021; Olsson et al., 2022). Moreover, we show that before induction heads form our transformer adopts simpler strategies, following a progression akin to that found by Olsson et al. (2022) for fully-developed models with increasing depth, or that found by Edelman et al. (2024) within the development of a single transformer in a Markovian sequence modeling setting.

For in-context linear regression, we see that before in-context learning emerges, the model predicts all outputs as roughly zero (the optimal context-independent prediction). Moreover, while the transformer can partially adapt to out-of-distribution regression tasks when in-context learning first emerges, this capability deteriorates in later stages. This deterioration is possibly an artifact of the "forgetting" phenomenon studied by Panwar et al. (2024).

**Development and model complexity.** While we have introduced the LLC as a measure of geometric degeneracy, it can also be understood as a measure of model complexity (cf. Appendix A.1). It is quite natural that certain changes in a model's internal computational structure would show up as a change in complexity. For example, Chen et al. (2024) showed that the emergence of a certain structure in a transformer coincided with a spike in two model complexity measures, namely the model's Fisher information and the intrinsic dimension (Facco et al., 2017) of the model's embeddings.

A noteworthy aspect of our experiments are stages in which the LLC *decreases,* corresponding to a *simplification* of the internal computational structure of the model. As an empirical phenomenon, such model simplification has precedent, for instance with Chen et al. (2024) and the recent literature on grokking (Power et al., 2022; Nanda et al., 2023; Notsawo et al., 2024). In our case, the mechanistic nature of the simplification is not yet clear, with the collapse of layer normalization, embedding, and attention patterns arising as candidates in the in-context linear regression setting.

This phenomenon is noteworthy because it is currently not accounted for by theories of neural network development. In the theory of saddle-to-saddle dynamics, deep linear networks learn progressively *more* complex approximations of the data (Saxe et al., 2019). The singular learning process as outlined in Section 4 also describes transitions for which the LLC increases, though decreasing the LLC while holding the loss constant would be another way to decrease the free energy according to equation (4). Providing a theoretical account of these developmental stages is an open problem.

## REPRODUCIBILITY STATEMENT

Detailed descriptions of our experimental setups, including model architectures, training procedures, and hyperparameters, are provided in Appendix D. For the language modeling experiments, specifics can be found in Appendix D.1, including model architecture (Appendix D.1.1), tokenization (Appendix D.1.2), and training details (Appendix D.1.3). For the in-context linear regression experiments, corresponding details are in Appendix D.2, covering model architecture (Appendix D.2.1), tokenization (Appendix D.2.2), and training procedures (Appendix D.2.3).

Our LLC estimation procedure is thoroughly documented in Appendix D.3, which provides a step-by-step guide for calibrating LLC estimates in novel settings. This includes discussions on varying temperature (Appendix D.3.1), seeding random noise (Appendix D.3.2), and calibrating key hyperparameters (Appendix D.3.3). Final hyperparameter values for the language modeling experiments and linear regression experiments are detailed in Appendices D.1.4 and D.2.4, respectively.

The metrics used for behavioral and structural analyses are detailed in Appendices B and C for language models and regression transformers, respectively. These sections provide precise definitions and implementation details for each metric.

To facilitate reproduction of our analyses, we have made our codebase available. The anonymized repository containing additional figures and code can be accessed at the URL provided in Appendix E.

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

# Appendix

**Appendix A** reviews the learning coefficient, providing some simple toy examples contrasting the learning coefficient and Hessian-based measures. This section also discusses various additional details involved in SGLD-based LLC estimation.

**Appendix B** and **Appendix C** examine the developmental stages of language and linear regression in more detail and explain the various metrics we use to track geometric, behavioral, and structural development.

**Appendix D** covers experimental details, such as model architecture, training procedure, and hyperparameters for LLC estimation. We end this section with a worked through treatment of the calibrations involved in applying LLC estimation to regression transformers to serve as a reference (Appendix D.3).

**Appendix E** contains a link to additional figures and the codebase used to run the experiments in this paper.

## A  THE LOCAL LEARNING COEFFICIENT (LLC)

### A.1  INTERPRETATIONS AND EXAMPLES OF THE LLC

In Section 3, we introduced the LLC as a quantification of geometric degeneracy. In this section, we discuss an additional perspectives on the LLC as a count of the "effective" dimensionality of a parameter, and we give additional examples of the LLC. We refer the reader to Watanabe (2009) and Lau et al. (2024) for more discussion.

The LLC has some similarity to an effective parameter count. If the population loss $\ell$ looks like a quadratic form near $w^*$ then $\lambda(w^*) = \frac{d}{2}$ is half the number of parameters, which we can think of as $d$ contributions of $\frac{1}{2}$ from every independent quadratic direction. If there are only $d-1$ independent quadratic directions, and one coordinate $w_i$ such that small variations in $w_i$ near $w_i^*$ do not change the model relative to the truth (this dimension is "unused") then $\lambda(w^*) = \frac{d-1}{2}$.

The situation becomes more intricate when certain dimensions are degenerate but not completely unused, varying to quartic or higher order near the parameter (rather than being quadratic or flat). While every unused coordinate reduces the LLC by $\frac{1}{2}$, changing the dependency on a coordinate from quadratic ($w_i^2$) to quartic ($w_i^4$) (increasing its *degeneracy* while still "using" it) reduces the contribution to the LLC from $\frac{1}{2}$ to $\frac{1}{4}$.

As a source of intuition, we provide several examples of exact LLCs:

- $\ell(w_1, w_2, w_3) = aw_1^2 + bw_2^2 + cw_3^2$ with $a, b, c > 0$. This function is nondegenerate, and $\lambda(0, 0, 0) = \frac{1}{2} + \frac{1}{2} + \frac{1}{2} = \frac{3}{2}$. This is independent of $a, b, c$. That is, the LLC $\lambda$ does *not measure curvature*. For this reason, it is better to avoid an intuition that centers on "basin broadness" since this tends to suggest that lowering $a, b, c$ should affect the LLC.
- $\ell(w_1, w_2, w_3) = w_1^2 + w_2^2 + 0$ in $\mathbb{R}^3$ is degenerate, but its level sets are still submanifolds and $\lambda(0, 0, 0) = \frac{1}{2} + \frac{1}{2}$. In this case the variable $w_3$ is unused, and so does not contribute to the LLC.
- $\ell(w_1, w_2, w_3) = w_1^2 + w_2^4 + w_3^4$ is degenerate and its level sets are, for our purposes, not submanifolds. The singular function germ $(\ell, 0)$ is an object of algebraic geometry, and the appropriate mathematical object is not a *manifold* or a *variety* but a *scheme*. The quartic terms contribute $\frac{1}{4}$ to the LLC, so that $\lambda(0, 0, 0) = \frac{1}{2} + \frac{1}{4} + \frac{1}{4} = 1$. The higher the power of a variable, the greater the degeneracy and the lower the LLC.

Figure 2 offers several additional examples, from left to right:

- A quadratic potential $\ell_1(w_1, w_2) = w_1^2 + w_2^2$, for which the LLC is maximal in two dimensions, $\lambda_1(0, 0) = d/2 = 1$.
- A quartic potential $\ell_2(w_1, w_2) = w_1^4 + w_2^4$, for which the LLC is $\lambda_2(0, 0) = 1/2$.

- An even more degenerate potential $\ell_3(w_1, w_2) = w_1^2 w_2^4$, for which $\lambda_3(0,0) = 1/4$. We note that Hessian-derived metrics cannot distinguish between this geometry and the preceding quartic geometry.
- A qualitatively distinct potential $\ell_4(w_1, w_2) = (w_1 - 1)^2(w_1^2 + w_2^2)^4$ from Lau et al. (2024) with the same LLC at the origin, $\lambda_4(0,0) = 1/4$.

While nondegenerate functions can be locally written as quadratic forms by the Morse Lemma (and are thus qualitatively similar to the approximation obtained from their Hessians), there is no simple equivalent for degenerate functions, such as the population losses of deep neural networks.

## A.2 Estimating LLCs with SGLD

We follow Lau et al. (2024) in using SGLD to estimate the expectation value of the loss in the estimator of the LLC. For a given choice of weights $w^*$ we sample $C$ independent chains with $T_{\text{SGLD}}$ steps per chain. Each chain $c$ is a sequence of weights $\{w_\tau^{(c)}\}_{\tau=1}^{T_{\text{SGLD}}}$. From these samples, we estimate the expectation $\mathbb{E}_{w|w^*,\gamma}^\beta[\mathcal{O}(w)]$ of an observable $\mathcal{O}$ by

$$\frac{1}{CT_{\text{SGLD}}} \sum_{c=1}^{C} \sum_{\tau=1}^{T_{\text{SGLD}}} \mathcal{O}(w_\tau^{(c)}), \tag{5}$$

with an optional burn-in period. Dropping the chain index $c$, each sample in a chain is generated according to:

$$w_{\tau+1} = w_\tau + \Delta w_\tau, \tag{6}$$
$$w_1 = w^*, \tag{7}$$

where the step $\Delta w_\tau$ comes from an SGLD update

$$\Delta w_\tau = \frac{\epsilon}{2}\left(\beta n \nabla \ell_m^{(\tau)}(w_\tau) + \frac{\gamma}{2}(w_\tau - w^*)\right) + \mathcal{N}(0, \epsilon). \tag{8}$$

In each step $\tau$ we sample a mini-batch of size $m$ and the associated empirical loss, denoted $\ell_m^{(\tau)}$, is used to compute the gradient in the SGLD update. We note that LLC estimator defined in (3) uses the expectation $\mathbb{E}^\beta[\ell_n(w)]$ which in the current notation means we should take $\mathcal{O}(w)$ to be $\ell_n(w)$. For computational efficiency we follow Lau et al. (2024) in recycling the mini-batch losses $\ell_m(w_\tau^{(c)})$ computed during the SGLD process. That is, we take $\mathcal{O} = \ell_m^{(\tau)}$ rather than $\mathcal{O} = \ell_n$.

More detailed results for language and regression are provided in Appendix B.2 and Appendix C.2, respectively. Appendix D.3 provides a walk-through for the LLC calibration process.

## A.3 LLC Estimates away from local minima

Our methodology for detecting stages is to apply LLC estimation to compute $\hat{\lambda}(w^*)$ at neural network parameters $w^* = w_t$ across training. In the typical case these parameters will *not* be local minima of the population loss, violating the theoretical conditions under which the LLC is defined.

It is not surprising that the estimator appears to work if $w^*$ is approximately a local minima. (Lau et al., 2024) validated their estimator at both parameters constructed to be local minima of the population loss and also at parameters found through training with stochastic gradient descent (possibly not local minima of the empirical loss, let alone the population loss). They showed that in both cases the estimator recovers the true learning coefficient associated with the global minimum of the population loss. On the other hand, if $w^*$ is far from any local minima, it is *a priori* quite surprising that the SGLD-based estimation procedure works at all, as in this situation one might expect the chains to explore directions in which the loss decreases.

Nevertheless, Chen et al. (2023) found that, empirically, LLC estimation away from local minima appears to give sensible results in practice. In our case, with sufficient localization we see stable estimates throughout training.

Theoretically accounting for this phenomenon is an interesting open problem. Perhaps there is a notion of *stably evolving equilibrium* in the setting of neural network training, echoing some of the

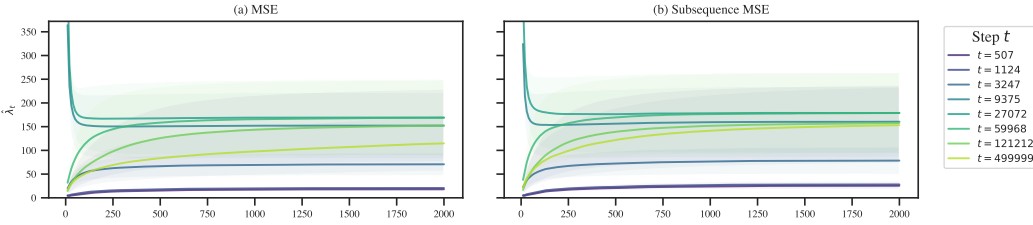

Figure A.1: Loss-based (*left*) and likelihood-based (*right*) LLC estimation yield identically ordered LLC estimates. With the exception of final checkpoint's LLC estimate (which is larger for the loss-based estimate), the values are close to identical. These plots display *LLC traces*, which show the LLC estimate as a function of *SGLD steps*. This is a useful tool for calibrating LLC estimation (Appendix D.3).

ideas of Waddington (1957), such that the LLC estimation procedure is effectively giving us the local learning coefficient of a different potential to the population loss—a potential for which the current parameter actually *is* at a critical point. We leave addressing this question to future work.

### A.4  LOG LIKELIHOOD-BASED LOSS

In the main body, we apply the LLC to empirical loss functions that do not arise as the log likelihood of independent random variables, due to the repeated use of dependent sub-sequences. Here we explain that it is possible to define a proper negative log likelihood over independent observations for the linear regression setting: similar observations can be made in the language modeling setting.

Let $\Pi(k)$ be a probability distribution over the context length $k$. Ideally, the transformer would be trained to make predictions $y_k$ given a context of length $k$ where $k$ is sampled from $\Pi$. With the given distribution over contexts this leads to a negative log likelihood of the form

$$L(w) = \sum_k p_k L_{[k]}(w) \tag{9}$$

where $p_k$ is the probability of sampling $k$ from $\Pi$ and

$$L_{[k]}(w) = \int q(S_k, y_k | \mathbf{t}, k) q(\mathbf{t}) \Big[ f_w(S_k) - y_k \Big]^2 dS_k \, dy_k \, d\mathbf{t} \tag{10}$$

using the notation of Section 2 so $S_k = (x_1, y_1, \dots, x_{k-1}, y_{k-1}, x_k)$ is a context of length $k$. It is straightforward to check that this negative log likelihood $L$ agrees with the population loss $\ell$ associated to the empirical loss defined in Section 2. However the empirical quantities $L_n(w)$ and $\ell_n(w)$ defined for a set of samples of size $n$ are *not* the same.

Since we use the empirical loss $\ell_n$ in our calculation of the estimated LLC, whereas the foundational theory of SLT is written in terms of the empirical negative log likelihood $L_n$, it is natural to wonder how much of a difference this makes in practice. Figure A.1 depicts LLC traces (Appendix D.3) for a highlighted number of checkpoints using either a likelihood-based estimate (with variable sequence length) or loss-based estimate (with fixed sequence length). The relative orderings of complexities does not change, and even the values of the LLC estimates do not make much of a difference, except at the final checkpoint, which has a higher value for the sub-sequence-based estimate.

### A.5  STAGE DISCOVERY

To identify stage boundaries, we look for plateaus in the LLC: checkpoints at which the slope of $\hat{\lambda}(w_t)$ over $t$ vanishes. To mitigate noise in the LLC estimates, we first fit a Gaussian process with some smoothing to the LLC-over-time curve. Then we numerically calculate the slope of this Gaussian process with respect to $\log t$. The logarithm corrects for the fact that the learning coefficient, like the loss, changes less as training progresses. We identify stage boundaries by looking for checkpoints at which this estimated slope equals zero. The results of this procedure are depicted in Figure B.2 for language and Figure C.1 for linear regression.

At a local minima or maxima of the estimated LLC curve identifying a plateau from this estimated slope is straightforward, since the derivative crosses the x-axis. However at a saddle point, the slope may not exactly reach zero, so we have to specify a "tolerance" for the absolute value of the derivative, below which we treat the boundary as an effective plateau.

In this case, we additionally require that the plateau be at a local minimum of the absolute first derivative. Otherwise, we may identify several adjacent points as all constituting a stage boundary.

To summarize, identifying stage boundaries is sensitive to the following choices: the intervals between checkpoints, the amount of smoothing, whether to differentiate with respect to $t$ or $\log t$, and the choice of tolerance. However, once a given choice of these hyperparameters is fixed, stages can be *automatically* identified, without further human judgment.

## B    DEVELOPMENTAL ANALYSIS OF LANGUAGE MODELS

In this section, we present further evidence on the geometric (Appendix B.2), behavioral (Appendix B.3), and structural (Appendix B.4) development of the language model over the course of training. In addition, we present results for a 1-layer attention-only model (Appendix B.5).

### B.1    UNIVERSALITY

Figure B.1a shows loss and LLC curves for five seeds (differing in model initialization and batch schedule). In each seed, LLC estimation reveals stage LM1–LM4. In three of the five seeds, stage LM5 is subdivided into two additional stages.

### B.2    GEOMETRIC DEVELOPMENT

Figure B.2 displays the test loss and LLC curves from Figure 1a in addition to the weight norm over time and associated slopes. Stage boundaries coincide with where the slope of the local learning coefficient crosses zero, that is, where there is a plateau in the LLC.

### B.3    BEHAVIORAL DEVELOPMENT

#### B.3.1    BIGRAM SCORE

We empirically estimate the conditional bigram distribution by counting instances of bigrams over the training data. From this, we obtain the conditional distribution $\tilde{q}(t'|t)$, the likelihood that a token $t'$ follows $t$. The *bigram score* $B_k^S$ at index $k$ of an input context $S$ is the cross entropy between the model's predictions $p(t_{k+1}|t_k)$ at that position and the empirical bigram distribution,

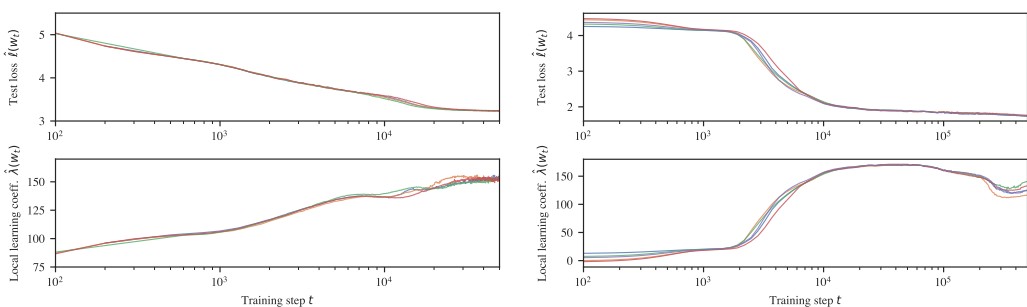

(a) **Two-layer attention-only language transformers.**     (b) **In-context linear regression transformers.**

Figure B.1: Figure 1a and Figure 1b for multiple seeds. In both settings, LLC reveals a consistent set of stages across five seeds. Late-training behavior shows more variance across seeds (see Appendix B.1 and Appendix C.1).

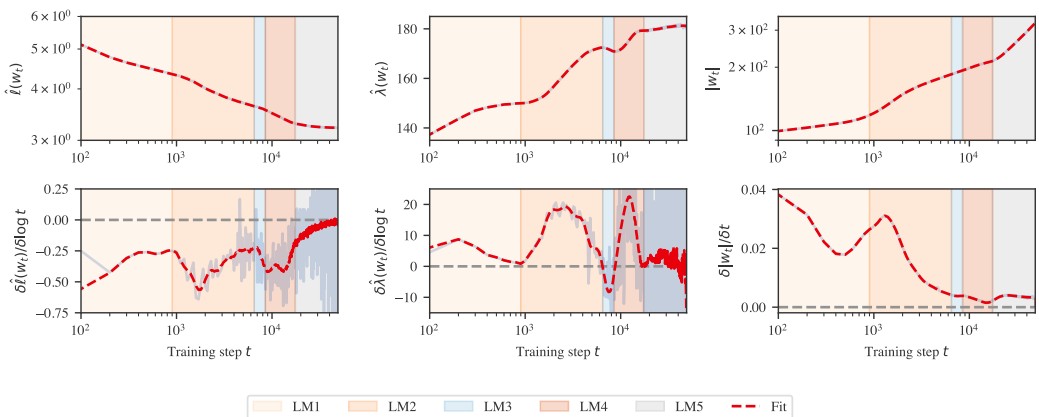

Figure B.2: A more detailed version of Figure 1a for two-layer language models. *Top*: Loss, LLC, and weight norm, along with an overlaid Gaussian process fit to these curves (red dotted lines). *Bottom*: Associated slopes, both numerically estimated finite differences (transparent blue) and of the Gaussian process (red dotted lined). Note that stage LM5 may be subdivided into further stages (Appendix A.5). However, the noise in LLC estimates late in training is high, so we do not draw any conclusions from this.

$$B_k^S = - \sum_{i=1}^{d_{\text{vocab}}} \tilde{q}(t_{k+1}^{(i)}|t_k) \log p(t_{k+1}^{(i)}|t_k), \tag{11}$$

where the $t_{k+1}^{(i)}$ range over the possible second tokens from the tokenizer vocabulary. From this we obtain the *average bigram score*

$$\bar{B} = \frac{1}{n} \sum_{i=1}^{n} B_{k_i}^{S_i}, \tag{12}$$

where we take fixed random sequences of $k_i$ and $S_i$ for $1 \le i \le n = 5,000$, which is displayed over training in Figure 4a. This is compared against the best-achievable bigram score, which is the bigram distribution entropy itself, averaged over the validation set.

### B.3.2  $n$-GRAM SCORES

In stage LM2 we consider $n$-grams, which are sequences of $n$ consecutive tokens, meaning 2-grams and bigrams are the same. Specifically, we consider *common* $n$-grams, which is defined heuristically by comparing our 5,000 vocab size tokenizer with the full GPT-2 tokenizer. We use the GPT-2 tokenizer as our heuristic because its vocabulary is constructed iteratively by merging the most frequent pairs of tokens.

We first tokenize the tokens in the full GPT-2 vocabulary to get a list of 50,257 $n$-grams for various $n$. The first 5,000 such $n$-grams are all 1-grams, after which 2-grams begin appearing, then 3-grams, 4-grams, and so on (where 2-grams and 3-grams may still continue to appear later in the vocabulary). We then define the set of common $n$-grams as the first 1,000 $n$-grams that appear in this list for a fixed $n$, $n \ge 2$.

If we track the performance on $n$-grams and see it improve, we may ask whether this is simply a function of the model learning to use more context in general, rather than specifically improving on the set of $n$-grams being tracked. We measure performance against this baseline by defining an $n$-*gram score*. For a fixed $n$, we obtain the average loss $\ell_{\text{gram}}^n$ of the model on predicting the final tokens of our set of 1,000 $n$-grams and also obtain the average loss $\ell_{\text{test}}^n$ of the model on a validation set at position $n$ of each validation sequence. The $n$-gram score is then defined to be $\ell_{\text{test}}^n / \ell_{\text{gram}}^n$.

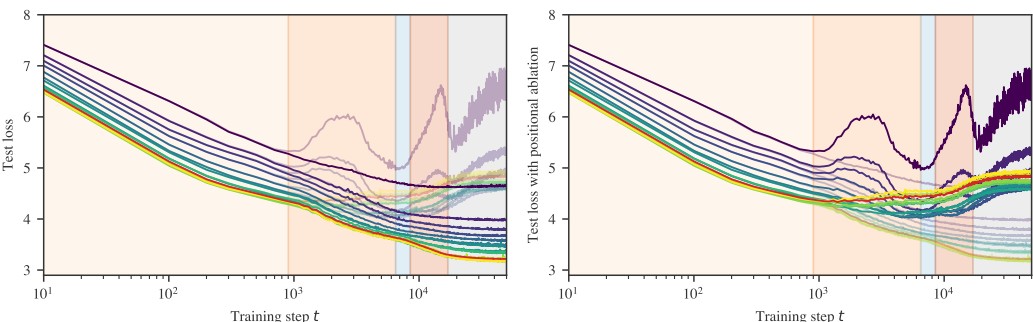

Figure B.3: The model learns to start using the positional encoding in LM2, when the performance starts to worsen when ablating the positional encoding. In both plots, earlier token positions are colored more purple, while later token positions are more yellow, and the overall mean loss is colored in red. Both sets of per-token losses are shown in both graphs for ease of comparison. *Left:* original test loss is emphasized. *Right:* test loss with the positional embedding ablated is emphasized.

### B.3.3 IN-CONTEXT LEARNING SCORE

The *in-context learning score* is a behavioral measure of the relative performance of a model later in a sequence versus earlier in the sequence. We follow a similar construction as Olsson et al. (2022), where we take the loss at the 500th token minus the loss at the 50th token, so that a more negative score indicates better performance later in the sequence. This is then averaged over a 100k-row validation dataset. The performance of the language model over the course of training can be seen at the bottom of Figure 4b.

### B.4 STRUCTURAL DEVELOPMENT

### B.4.1 POSITIONAL EMBEDDING

In Figure B.3, we measure the effect of the positional embedding on model performance by comparing the model's performance at particular context positions on a validation set over the course of training against performance on the same validation set but with the positional embedding zero-ablated. The full context length is 1024, and we measure test loss at positions 1, 2, 3, 5, 10, 20, 30, 50, 100, 200, 300, 500, and 1000. In the transition from stage LM1 to LM2, the model begins using the learnable positional embedding to improve performance. The difference between test loss with and without the positional ablation is negligible at all measured positions until the LM1–LM2 boundary.

Structurally, we might predict that the positional embeddings should organize themselves in a particular way: in order to understand relative positions, adjacent positions should be embedded close to each other, and far-away positions should be embedded far apart.

In Figure B.4, we examine the development of the positional embedding itself over time from two angles. The first is to take the embeddings of each position in the context and to run PCA on those embeddings. The result is that as training progresses, the positional embedding PCAs gradually resolve into Lissajous curves, suggesting that the positional embeddings might look like a random walk (Antognini and Sohl-Dickstein, 2018; Shinn, 2023). However, if we look to the explained variance, we see that it grows very large for PC1, reaching $94.2\%$ at training step 6400. This is much higher than we would expect for Brownian motion, where we expect to see about $61\%$ explained variance in PC1 (Antognini and Sohl-Dickstein, 2018).

The second perspective we use is to look at how the magnitudes of positional embeddings over the context length develop. In this case, we observe that the magnitudes seem to have a fairly regular structure. In conjunction with the PCAs and explained variance, we might infer that the positional embeddings look approximately like a (possibly curved) line in $d_{\text{model}} = 256$ dimensional space. A positional embedding organized in this way would make it easier for an attention head to attend to multiple recent tokens, which is necessary if a single head is to learn $n$-grams.

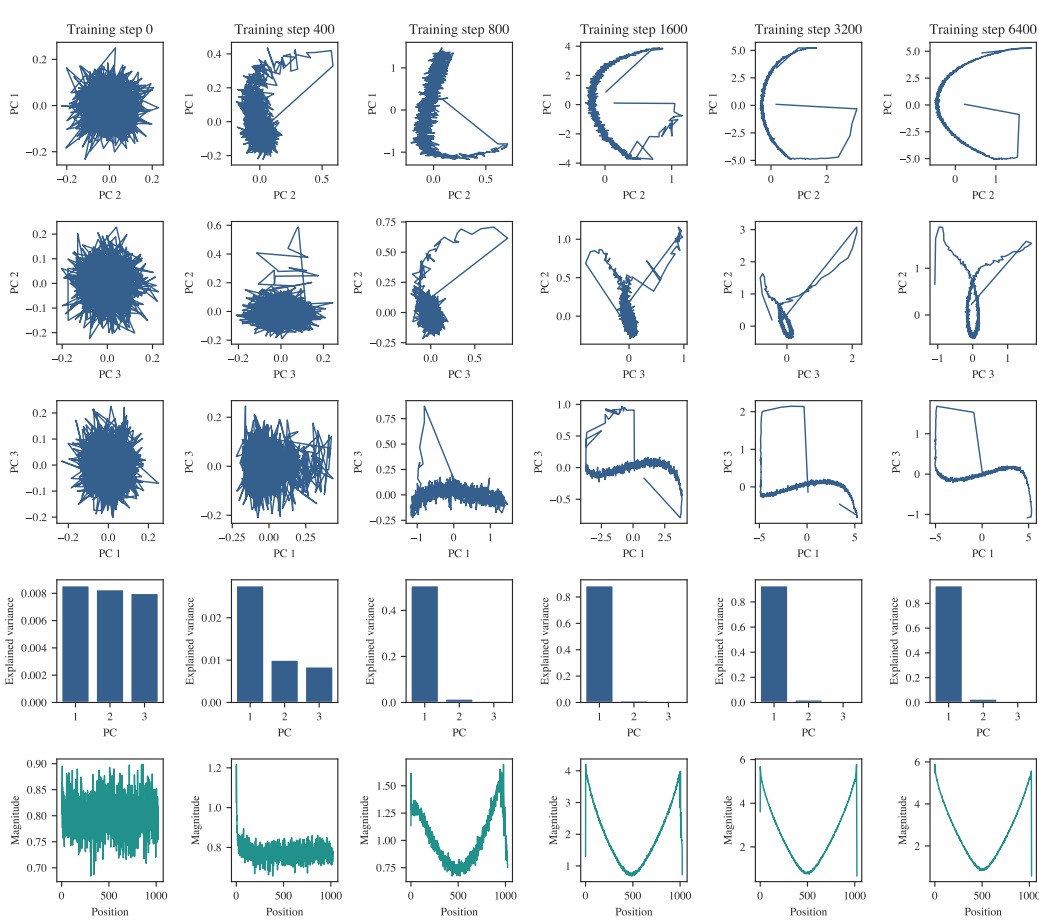

Figure B.4: Columns progress through training time at training steps 0, 400, 800, 1600, 3200, and 6400. The first three rows are plots of the first three principle components of PCA on the positional embedding weights, while the fourth row shows the explained variance for each of the principal components. The fifth row plots the magnitude of the embedding of each position in the context length of 1024.

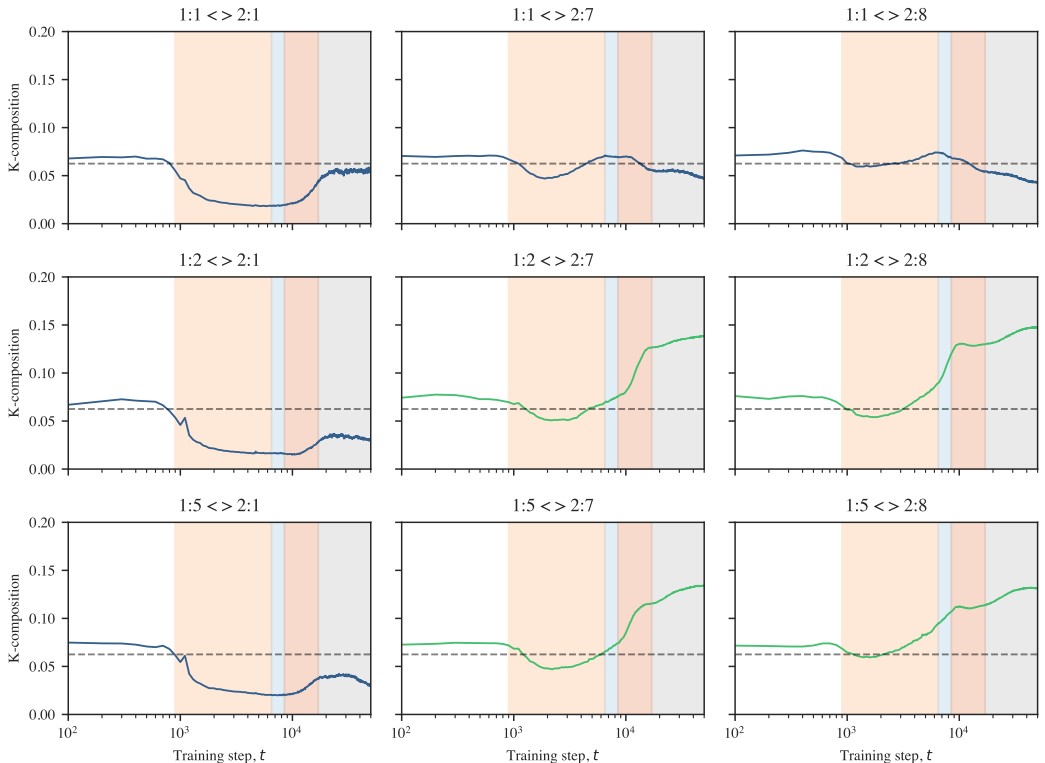

Figure B.5: The K-composition scores (Elhage et al., 2021) between first and second layer attention heads. The $h$th attention head in layer $l$ is indexed by $l : h$. The attention heads that eventually become previous token heads are $h = 2, 5$ in layer 1 (subplot rows 2 and 3), and the attention heads that eventually become induction heads are $h = 7, 8$ in layer 2 (subplot columns 2 and 3). The attention heads $1 : 1$ and $2 : 1$ are included for comparison. The induction heads $2 : 7$ and $2 : 8$ begin K-composing with first layer heads near the start of stage LM2. They continue to compose with the previous token heads in stages LM3 and LM4 (highlighted in green) while their K-composition scores drop with other attention heads in layer 1 in later stages.

### B.4.2 Composition scores

Let $W_Q^h, W_K^h, W_V^h$ be the query, key, and value weights of attention head $h$ respectively. There are three types of composition between attention heads in transformer models in Elhage et al. (2021):

- Q-Composition: the query matrix $W_Q^h$ of an attention head reads in a subspace affected by a previous head

- K-Composition: the key matrix $W_K^h$ of an attention head reads in a subspace affected by a previous head

- V-Composition: the value matrix $W_V^h$ of an attention head reads in a subspace affected by a previous head

If $W_O^h$ is the output matrix of an attention head, then $W_{QK}^h = W_Q^{h\ T} W_K^h$ and $W_{OV}^h = W_O^h W_V^h$. The composition scores are

$$||MW_{OV}^{h1}||_F / (||M||_F ||W_{OV}^{h_1}||_F) \tag{13}$$

Where $M = W_{QK}^{h_2\ T}$, $M = W_{QK}^{h_2}$, and $M = W_{OV}^{h_2}$ for Q-, K-, and V-Composition respectively. See Figure B.5 for K-composition scores over time between attention heads in the induction circuits.

### B.4.3 PREVIOUS-TOKEN MATCHING SCORE

The *previous-token matching score* is a structural measure of induction head attention. It is the attention score given to $[A]$ by an attention head at $[B]$ in the sequence $\ldots [A][B]$ (i.e., how much the head attends to the immediately preceding token).

We compute this score using a synthetic data generating process, generating 10k fixed random sequences with length between 16 and 64. The first token is a special "beginning of string" token, and the remaining tokens are uniformly randomly sampled from other tokens in the vocabulary.

For each sample in this synthetic dataset, we measure the attention score that an attention head gives to the previous token when at the last token in the sequence. These scores are averaged across the dataset to produce the previous-token matching score for that attention head at a given checkpoint. The progression of previous-token matching scores over time can be seen in Figure 4b.

### B.4.4 PREFIX MATCHING SCORE

The *prefix matching score* from Olsson et al. (2022) is defined similarly to the previous-token matching score. Given a sequence $[A][B] \ldots [A]$, the prefix matching score of a particular attention head is how much the attention head attends back to the first instance of $[A]$ when at the second instance of $[A]$.

We compute this score using a synthetic data-generating process. We first generate 10k fixed random sequences of length 128. The first token is always a special "beginning of string" token and the $[A]$ and $[B]$ tokens are selected and placed randomly. One $[A]$ token is placed in the first half of the sequence, the other is placed in the second half, and the $[B]$ token is placed directly after the first $[A]$ token. The remaining tokens are randomly sampled from the tokenizer vocabulary, excluding the $[A]$, $[B]$, and beginning of string tokens.

For each sample in this synthetic dataset, we measure the attention score that each attention head assigns to the earlier instance of $[A]$ from the latter instance of $[A]$. These scores are averaged across the dataset to produce the prefix matching score for that attention head at a given checkpoint. The progression of prefix matching scores over time can be seen in Figure 4b.

### B.5 ONE-LAYER MODEL RESULTS

We also trained and ran some experiments on a one-layer language model (see Appendix D.1.1 for details). We aggregate results for the one-layer language model here, mirroring the experiments for the two-layer language model where possible. The early development of the one-layer model has many parallels with the two-layer model. At a single stage boundary, just as it occurs in the two-layer model, the one-layer model minimizes its bigram score (see Appendix B.3.1), begins utilizing the positional embedding to noticeably improve performance (see Appendix B.4.1), and starts making sudden improvements to the same $n$-gram scores (see Appendix B.3.2). Remarkably this occurs at the same checkpoint as in the 2-layer model (at 900 training steps).

One key difference, however, is that this occurs at the *second* stage boundary as discerned by the plateaus of the LLC estimation. We did not closely investigate why the LLC estimation appears to drop between steps 400 and 900 in this model. As a result though, we do observe an interesting qualitative similarity to the drop in LLC in stage LM3 of the two-layer model, that this drop precedes a noticeable bump in the loss function.

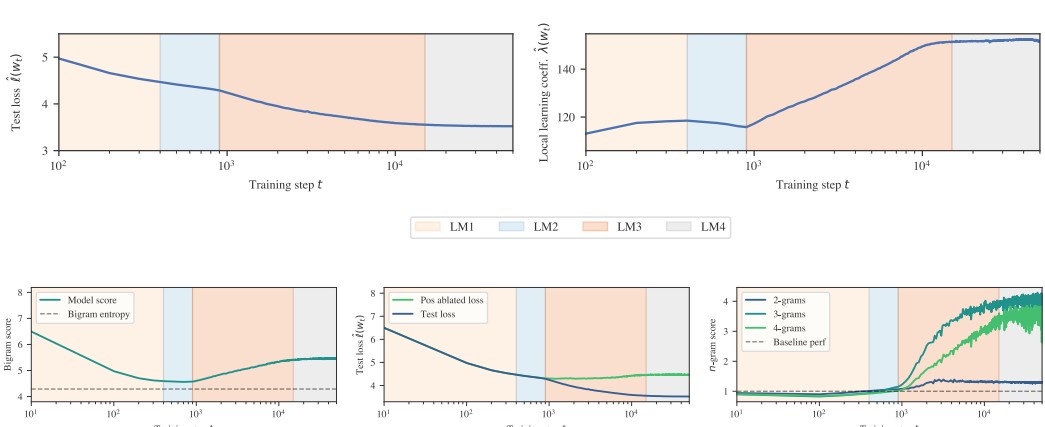

Figure B.6: We train a one-layer transformer model in the language setting to compare with the two-layer model. The development of certain behavioral and structural metrics over time closely mirrors the development of the same metrics in the early stages of the two-layer language model. *Top:* test loss and LLC estimations over time for the one-layer attention-only transformer, compare with Figure 1a. *Bottom:* bigram score, test loss with positional embedding ablated, and $n$-gram scores for the one-layer attention-only transformer, compare with Figure 4a.

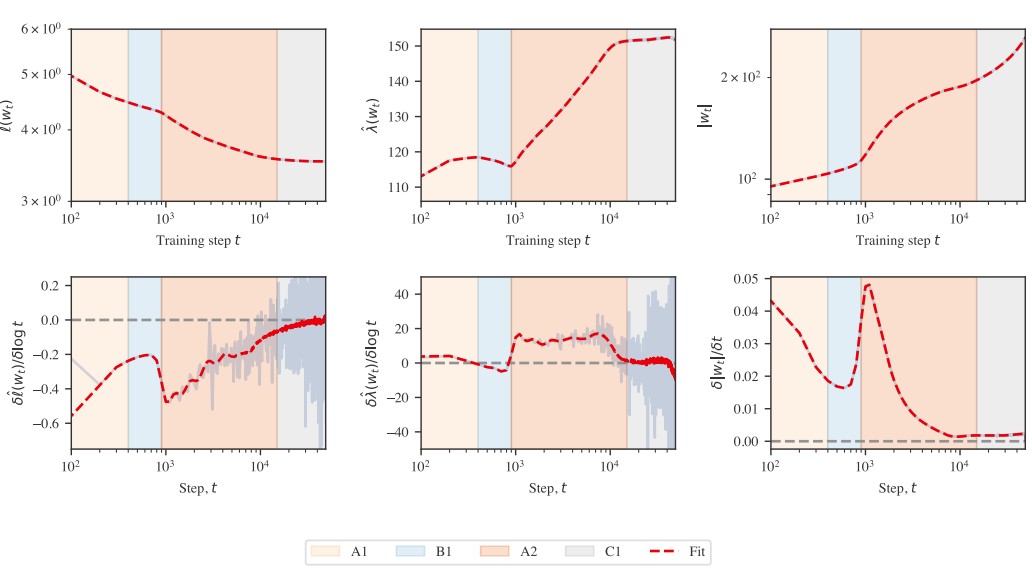

Figure B.7: A more detailed version of Figure B.6 for the one-layer language model. *Top*: Loss, LLC, and weight norm, along with an overlaid Gaussian process fit to these curves (red dotted lines). *Bottom*: Associated slopes, both numerically estimated finite differences (transparent blue) and of the Gaussian process (red dotted lined).

## C  DEVELOPMENTAL ANALYSIS OF REGRESSION TRANSFORMERS

In this section, we present further evidence on the geometric (Appendix C.2), behavioral (Appendix C.3), and structural (Appendix C.4) development of the transformer in the setting of in-context linear regression. In particular, we contrast the use of LLC with Hessian-derived statistics in Appendix C.2.2.

### C.1  UNIVERSALITY

Figure B.1b shows loss and LLC curves for five unique seeds (differing in model initialization and batch schedule). In each seed, LLC estimation reveals stages LR1–LR5. There is remarkably little variance across different seeds.

### C.2  GEOMETRIC DEVELOPMENT

#### C.2.1  LLC

Figure C.1 displays the test loss and LLC curves from Figure 1b in addition to the weight norm over time, and numerically estimated slopes associated to these three metrics. As in the case of language models, we identify stage boundaries by looking for plateaus in the local learning coefficient. Unlike the language models, here the boundaries LR1–LR2 and LR2–LR3 are clearly visible in the loss.

#### C.2.2  HESSIANS

Figure 6b shows the curvature-based notion of flatness captured by the Hessian (in contrast to the degeneracy-based notion of flatness captured by the LLC). To estimate the trace and maximum eigenvalues shown in this figure, we use the PyHessian library (Yao et al., 2020) over a batch of $m = 1024$ samples.

Crucially, we observe that these Hessian-derived metrics (henceforth, "curvature") and the LLC are *not* consistently correlated. During the first part of LR2, the LLC and the curvature are jointly increasing. Starting at around $t = 20\text{k}$, while the LLC is still increasing, the curvature starts decreasing. In the first part of LR3, both metrics decrease in tandem, but as of around $t = 120\text{k}$, the curvature turns around and starts increasing.

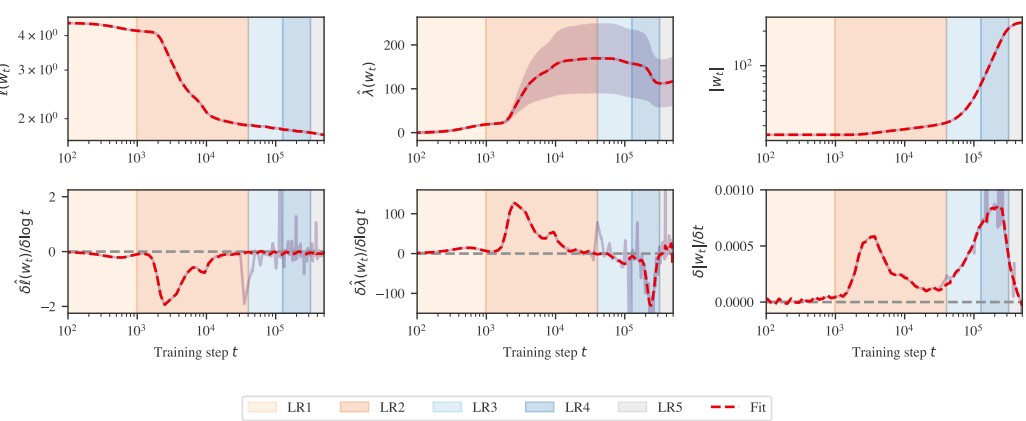

Figure C.1: A more detailed version of Figure 1b for linear regression. *Top*: Loss, LLC, and weight norm, along with an overlaid Gaussian process fit to these curves (red dotted lines). *Bottom*: Associated slopes, both numerically estimated finite differences (transparent blue) and of the Gaussian process (red dotted lined). *Top middle*: Error bars displaying the standard deviation over the 10 SGLD chains are displayed in the background. Note that large error bars across chains are to be expected. Between different SGLD estimations, the variance is much lower. For example, averaged over training, the standard deviation over different seeds is only 4.2.

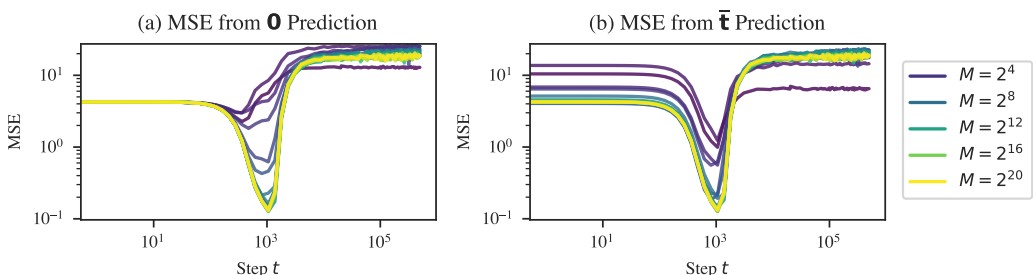

Figure C.2: **Learning the task prior** is universal across models trained on very different data distributions. Each line represents a model trained on a data distribution with a different number of $M$ distinct tasks ("task diversity" in Raventós et al. (2023)). In addition to taking a finite $M$, the models depicted here differ from the other models considered in this paper in that the former were trained with a maximum learning rate of $0.01$, and the models (inadvertently) lack an output matrix after the multi-head attention layer.

The Hessian fails to detect three of the four stage boundaries. Since these Hessian-based metrics are dominated by the largest eigenvalues — the directions of maximum curvature — they fail to observe the finer-grained measures of degeneracy that dominate the LLC. Moreover, we observe that LLC estimation is more scalable (empirically, it seems to be roughly linear in parameter count) than estimating the full Hessian (which is quadratic).

## C.3 BEHAVIORAL DEVELOPMENT

### C.3.1 TASK PRIOR SCORE

In addition to training models on a data distribution in which tasks $\mathbf{t}$ are generated on-the-fly, we examine the setting of Raventós et al. (2023), in which a finite set of $M$ tasks is generated ahead of time, and training samples involve randomly selected tasks from this set.

Figure C.2 depicts (a) the mean square distance between the model's predictions and the zero prediction in addition to (b) the mean square distance between the model's predictions and the "task prior" prediction, using the component-wise averaged $\bar{\mathbf{t}}$ over the set of tasks encountered during training. For all models, the minimum distance to the task prior prediction is lower than the minimum distance to the zero prediction. Hence, we call stage LR1 "learning the task prior" rather than simply learning the zero prediction.

### C.3.2 ICL

We consider two variants of the ICL score: $\mathrm{ICL}_{1:D}$, and $\mathrm{ICL}_{D:K}$.

If the noise term $\sigma^2$ equals zero and both tasks $\mathbf{t}$ and inputs $x_k$ are normalized (i.e., $\mathbf{t} \in S^{D-1}$), then $D - 1$ observations of input-output pairs are enough to precisely identify $\mathbf{t}$. Therefore, $\mathrm{ICL}_{1:D}$ measures how successful the model is at initially locating the task. The fact that the tasks and inputs are not normalized changes this only slightly: the task will still sit near $S^{D-1}$ within a shell of vanishing thickness as $D \to \infty$.

Once localized, $\mathrm{ICL}_{D:K}$ measures how successfully the model refines its internal estimate of $\mathbf{t}$ with additional examples, which it can use to reduce the error due to noise.

In terms of implementation, it's not necessary for the model to internally make a distinction between locating and refining its estimate of the task. For example, ridge regression makes no distinction. Still, we find it useful for reasoning about the progression of the model. In particular, we note that early in stage LR2, while the model begins to develop ICL for early tokens, it becomes *worse* at ICL over tokens late in the context. Later, at around 23k steps, $\mathrm{ICL}_{D:K}$ stabilizes, while $\mathrm{ICL}_{1:D}$ continues improving over the entire training run.

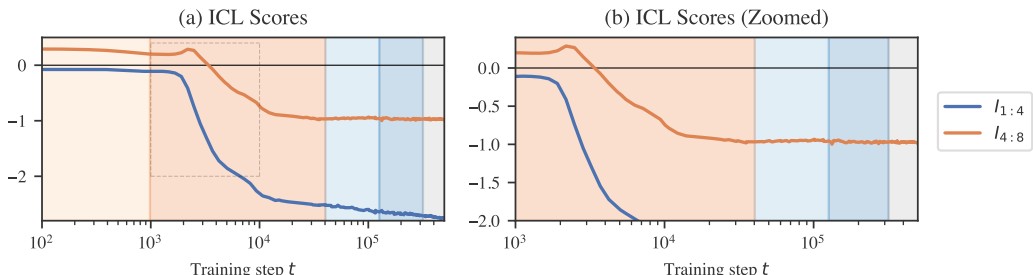

Figure C.3: **ICL scores** for the linear regression model. *Right*: ICL scores between inputs 1 and 4 and inputs 4 and 8 over time. We see that ICL emerges during the first half of LR2. *Left*: Highlighted ICL score curves from the end of LR1 to halfway through LR2. Note that when the model first starts improving on early tokens, it temporarily becomes worse at predicting later tokens. Note also that the model ceases to become better at later tokens as of the second half of LR2, whereas ICL on early tokens continues to improve throughout training.

### C.3.3 OOD GENERALIZATION

To further investigate behavior in stages LR2 and LR3, we probe the model on data sampled from different distributions than encountered during training.[2] We evaluate behavior on two families of perturbations: "OOD inputs" $x_k$, sampled according to a different scale

$$x_k \sim \mathcal{N}(0, gI_D), \tag{14}$$

for some gain parameter $g$, and "OOD tasks"

$$\mathbf{t} \sim \mathcal{N}(0, gI_D). \tag{15}$$

Note that these inputs and tasks are not out-of-distribution in the sense of coming from a distribution with a different support than the training distribution. However, the samples drawn from these "extreme" distributions are exponentially suppressed by the original training distribution.

Between $t = 1$k and $t = 4$k the model's outputs rapidly *diminish* in scale for out-of-distribution samples, both for $g > 1$ and $g < 1$, especially for out-of-distribution *inputs*. While the model is moving away from predicting with the task prior for in-distribution samples, it moves closer to predicting with the task prior for-in-distribution samples.

Between $t = 4$k and $t = 23$k, the model recovers on moderately out-of-distribution inputs $g < 10^{1.5}$ with performance remaining close to constant beyond this range. Past this stage, performance improves constantly for out-of-distribution tasks.

For out-of-distribution inputs, performance eventually worsens for some ranges of $g$. Between $t = 23$k and $t = 80$k the model further approaches the task prior prediction for extreme out-of-distribution inputs $g > 10^{1.5}$. Subsequently, between $t = 75$k and $t = 130$k the model moves away from the task prior prediction for extreme inputs, and performance deteriorates for inputs with $g > 10^{0.5}$. As of LR5, performance is roughly constant.

### C.4 STRUCTURAL DEVELOPMENT

### C.4.1 EMBEDDING

The embedding matrix $W_E$ is a linear transformation from $\mathbb{R}^{D+1} \to \mathbb{R}^{d_{\text{embed}}}$. Plotting the $D + 1$ singular values of this matrix, we notice that the embedding partially loses one of its components starting at the end of LR2 (Figure C.5a).

The input "tokens" $x_k$ span a $D$-dimensional subspace of the $D + 1$-dimensional "token space." The target tokens $y_k$ span an orthogonal 1-dimensional subspace. The collapse of one of the embedding

---

[2]Cf. Raventós et al. (2023) evaluating models trained on a set of discrete tasks on the "true" distribution consisting of novel tasks.

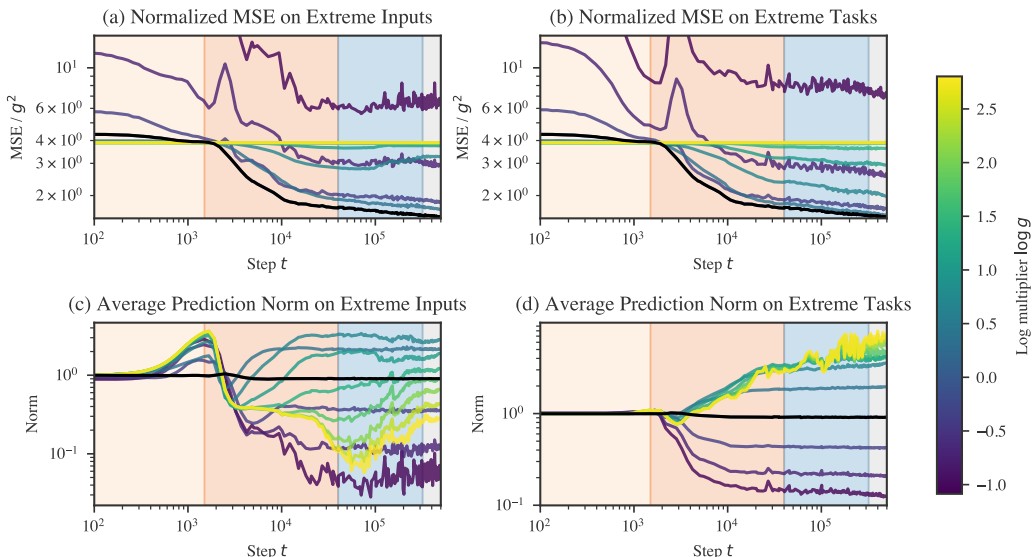

Figure C.4: Performance on extreme inputs over time may reveal additional substages in LR2 and in LR3. *Left*: The model first becomes better, then worsens at ICL on inputs sampled from $\mathcal{N}(0, gI_D)$ for large $g$. *Right*: The model continues to improve on ICL at tasks sampled from $\mathcal{N}(0, gI_D)$. *Top*: Normalized loss (divided by $g^2$) over time for OOD inputs and tasks. *Bottom*: Average $|\hat{y}|$ over time for OOD inputs and tasks.

matrix's singular values means that the model learns to redundantly encode the inputs and targets in the same $D$-dimensional subspace of the space of residual stream activations. The almost order of magnitude separation in the magnitudes of the square singular value means that the $D + 1$-th component of the token embedding explains only 2.9% of the variance in activations of the residual stream immediately after the embedding, whereas the dominant components explain roughly 24% each.

**Contributions to degeneracy.** Given a linear transformation $T_1 : \mathbb{R}^{D_1} \to \mathbb{R}^{D_2}$ followed by another linear transformation $T_2 : \mathbb{R}^{D_2} \to \mathbb{R}^{D_3}$, reducing the rank of $T_1$ from $r$ to $r' < r$ renders $D_3(r - r')$ components of the second transformation irrelevant. This would mean a *decrease* in the learning coefficient of $D_3(r - r')/2$ (a decrease in the *effective dimensionality* of $d$ leads to a decrease in the LLC of $d/2$[3]). In the actual model, we don't see an exact decrease in the rank, and a layer normalization sits between the linear transformation of the embedding and the linear transformations of each transformer block and unembedding. It is unclear what the precise relation between structure and geometry is in this case (Appendix C.4.6). Still, suggestively, the onset of embedding collapse coincides with a decrease in the rate of increase of $\hat{\lambda}(w_t)$.

### C.4.2 POSITIONAL ENCODING

The positional encoding goes through a similar collapse to the unembedding starting during the second part of LR2 and continuing into LR3 (Figure C.5b). Additionally, throughout these stages, the subspace spanned by the embedding becomes more aligned with the subspace spanned by the positional encoding (Figure C.5c).

**Contributions to degeneracy.** For the same reason as with the token embedding, a decrease in the dimensionality of the subspace occupied by activations reduces the effective number of dimensions and thus the learning coefficient. This occurs both as the positional encoding's effective

---

[3]Note that this is not the only possible way for the LLC to decrease. Changing the local loss landscape from quadratic to quartic or some higher power would also lower the LLC, by a fractional amount.

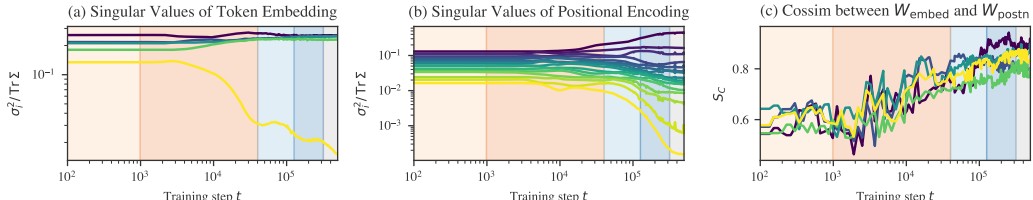

Figure C.5: *Left*: The embedding partially "collapses" during the second half of LR2. At the start of stage LR2, the minimum singular values explains only 3% of the variance in residual stream activations due to the sample. By the end of training, it explains half that. *Middle*: The positional encoding goes through a similar shift during LR3 (that begins earlier during LR2). *Right*: The cosine similarity between the 5 rows of $W_{\text{embed}}$ and the projection of those rows onto the subspace spanned by $W_{\text{unembed}}$ shows that the model learns to write to the same write tokens and positional information to the same subspace.

dimensionality decreases (vanishing singular values, Figure C.5b) and as the token embedding subspace and positional embedding subspace align (increasing cosine similarity, Figure C.5b).

### C.4.3 UNEMBEDDING COLLAPSE

The unembedding block consists of a layer normalization layer $\text{LN}(z)$ followed by a linear transformation $W_U z + b_U$ and finally a projection $\pi_y$ to extract the $y$-component. Given the 64-dimensional vector of activations $z$ in the residual stream right before the unembedding (for a specific token), the full unembedding operation is:

$$\pi_y \left[ W_U \left( \frac{z - \mathbb{E}[z]}{\sqrt{\mathbb{V}[z] + \epsilon}} \odot \gamma + \beta \right) + b_U \right]$$

where $\odot$ denotes element-wise multiplication of two vectors and $\gamma, \beta$ are the layer normalization weights and biases respectively.

**Effective unembedding weights and biases.** Moving terms around, we can represent this as

$$\left( (W_U)_{[0,:]} \odot \gamma \right) \left( \frac{z - \mathbb{E}[z]}{\sqrt{\mathbb{V}[z] + \epsilon}} \right) + \left( (W_U)_{[0,:]} \beta \right) + (b_U)_{[0]}$$

where we order the outputs so that the $y$-token corresponds to the 0th row. Because we are reading out a single $y$ component, we can express the unembedding transformation in terms of "effective" unembedding weights and biases

$$\tilde{W}_U = (W_U)_{[0,:]} \odot \gamma,$$
$$\tilde{b}_U = \left( (W_U)_{[0,:]} \beta \right) + (b_U)_{[0]}.$$

**Unembedding weights over time.** In Figure C.6, we plot $(\gamma, \beta)$, $((W_U)_{[0,:]}, (b_U)_{[0]})$, and $(\tilde{W}_U, \tilde{b}_U)$ as a function of training steps, along with the mean weight over time. These are 64- and 1-dimensional vectors, so we can display the entire set of components. During stage LR3 the majority of weights $\beta$ and $W_U$ "collapse" to zero. Additionally, the layer normalization biases temporarily experience a large increase in variance before returning to small values. Despite this, the mean of the linear weights, layer normalization biases, and effective weights remains remarkably constant and close to zero throughout the entire process.

**Contributions to degeneracy.** Suppose that $D$ of the layer normalization weights have vanished, say $\gamma_i = 0$ for $1 \le i \le D$. Then the corresponding columns of $W_U$ only contribute to the unembedding via their product $(W_U)_{[:,1:D]} \beta_{[1:D]}$ with the first $D$ rows of $\beta$. This creates a typical form of degeneracy studied in SLT and found, for example, in deep linear networks, where we can change the weights to $(W_U)_{[:,1:D]} A, A^{-1} \beta_{[1:D]}$ for any invertible $D \times D$ matrix $A$ without changing the function computed by the network. If in addition the $\beta_i$ vanish for $1 \le i \le D$ then the entries of $(W_U)_{[:,1:D]}$ are completely unconstrained, creating further degeneracy.

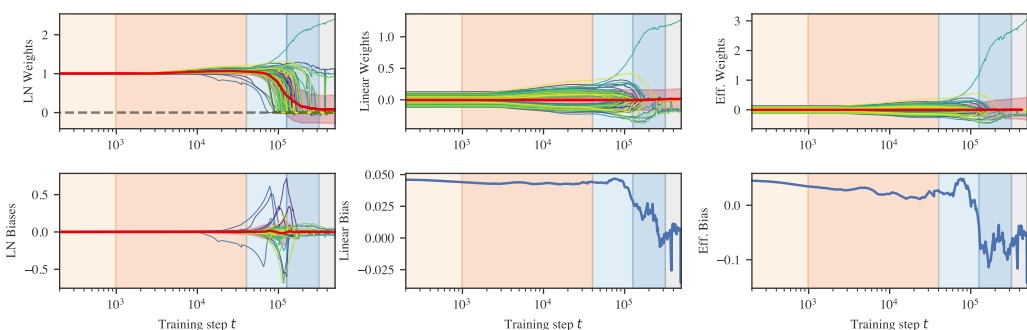

Figure C.6: **Unembedding weights over time** for the RT 1 transformer undergo a "collapse" that begins towards the end of LR2. When these weights reach zero in LR3 and LR4, it may contribute to the observed decrease in the LLC. *Top*: Weights over time. The outlier in the positive direction is the weight for the $y$-token output. *Bottom*: Biases over time. *Left*: Unembedding layer normalization weights over time. *Middle*: Unembedding linear weights over time (restricted to $y$-subspace). *Right*: Effective unembedding weights over time (obtained by element-wise multiplication of preceding columns, and focusing on the bias for only the $y$-token.

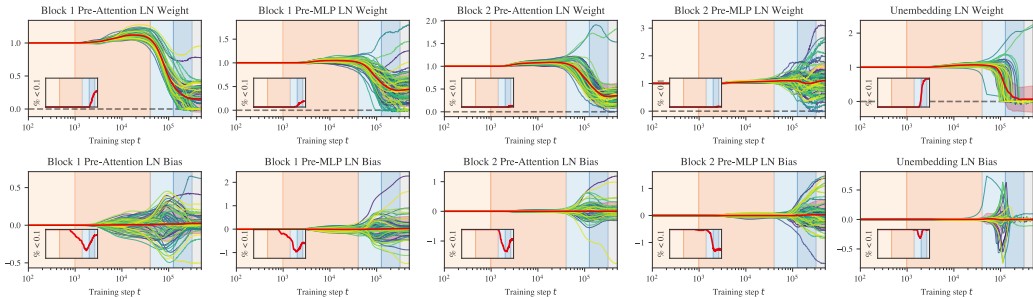

Figure C.7: **Layer norm weights over time**. *Top*: After LR3, the layer normalization collapse expands from the unembedding to earlier layers, most notably in the first pre-attention layer norm. This occurs without explicit regularization and may contribute to the concurrent decrease in LLC. *Bottom*: During layer normalization collapse, the variance of layer normalization biases increases drastically while the mean of the biases remains relatively constant. *Inset*: Plotting the fraction of weights or biases whose magnitude is less than 0.1 over time reveals that the collapse is more measured for intermediate layer norms: weights shrink to small values but not extremely close to zero as in the unembedding and first attention layer.

### C.4.4 LAYER NORMALIZATION COLLAPSE

The "collapse" in layer normalization weights is not unique to the unembedding. As depicted in Figure C.7, this behavior occurs in all layer norms except for the second MLP. The biases also remain centered close to zero even as the variance in biases grows much larger. Unlike in the unembedding, these layers begin to change earlier (starting halfway through LR2).

What is most striking about the layer normalization collapse is that it occurs without any explicit regularization (neither weight decay nor dropout). As such, it demonstrates a clear example of *implicit regularization*, i.e., inductive biases in the optimizer or model that favor simpler solutions.

**Contributions to degeneracy.** In the previous section, we describe how layer norm collapse in the unembedding is linked to an increase in degeneracy because it ensures that parameters in the subsequent linear layer become irrelevant. The same is true for layer norm which precedes the attention and MLP blocks.

### C.4.5 ATTENTION COLLAPSE

Over the course of training, we observe that some attention heads learn to attend *solely* (soft attention becomes hard attention) and *consistently* to certain positions (the attention pattern becomes content-independent). We call this phenomenon *attention collapse* in parallel with the other observed forms of collapse. Not only does this potentially contribute to a decrease in the LLC, but it also makes the attention heads identifiable: we find a self-attention head, previous-attention heads, previous-$x$-attention heads, and previous-$y$-attention heads.

$x$**-attention vs. $y$-attention.** For convenience we separate each attention head in two: one part for the $x$-tokens, and the other for the $y$-tokens.

**Attention entropy score.** To quantify attention hardness, we use the *attention entropy score* (Ghader and Monz, 2017; Vig and Belinkov, 2019). Given the attention pattern $\alpha_{k,k'}^{(b,h)}$ for how much token $k$ in head $h$ in block $b$ attends back to token $k'$, its attention entropy score $H_k^{(b,h)}$ is the Shannon entropy over preceding indices $k' < k$,

$$H_k^{(b,h)} = - \sum_{k' \leq k} \alpha_{k,k'}^{(b,h)} \log_2 \alpha_{k,k'}^{(b,h)}. \tag{16}$$

From this, we compute the normalized entropy $\hat{H}_k^{(b,k)}$, which divides the attention entropy by the maximum entropy for the given context length,

$$\hat{H}_k^{(b,h)} = \frac{H_k^{(b,h)}}{\log_2(k)}. \tag{17}$$

This accounts for the entropy being calculated over different numbers of tokens and is displayed in Figure C.8. Notably, the identified stages line up closely to stages of these attention entropy curves.

**Constant attention.** Accomplishing constant attention requires the presence of biases in the query and key transformations, or if there is no bias (as is the case for the models we investigated), requires attending to the positional embedding. With the Shortformer-style positional encoding used for the language models (Appendix D.1.1), this is straightforward: the positional information is injected directly into the key and weight matrices. With the linear regression models, where the positional embedding is added to the residual stream activations, this is less straightforward: achieving constant attention requires separating residual stream activations into orthogonal positional- and input-dependent subspaces, then reading from the former with the query and key weight matrices.

**Attention variability score.** To quantify how constant the attention pattern is, we use measure *attention variability* (Vig and Belinkov, 2019),

$$V_k^{(b,h)} = \frac{\sum_{i=1}^n \sum_{k' \leq k} \left| \alpha_{k,k'}^{(b,h)}(S_K^{(i)}) - \bar{\alpha}_{k,k'}^{(b,h)} \right|}{2n \sum_{k' \leq k} \bar{\alpha}_{k,k'}^{(b,h)}}, \tag{18}$$

where the division by 2 ensures the variability lies in the range $[0, 1]$. This is displayed in Figure C.9. These reveal that though attention hardness and variability are independent axes of differentiation, empirically, we observe that hard attention is correlated with low variability.

**Self-attention score.** Self-attention is measured by the average amount a token $k$ attends to itself, $\alpha_{k,k}^{(b,h)}$.

**Previous-token attention score.** Previous-token attention is measured the same as in the language model setting (Appendix B.4) with one difference: we compute the previous-token score not over a synthetic dataset but over a validation batch.

$x$**-attention score.** The total amount attended to inputs $x_k$, that is $\alpha_{k,x}^{(b,h)} = \sum_{k'=1}^K \alpha_{k,2k}^{(b,h)}$.

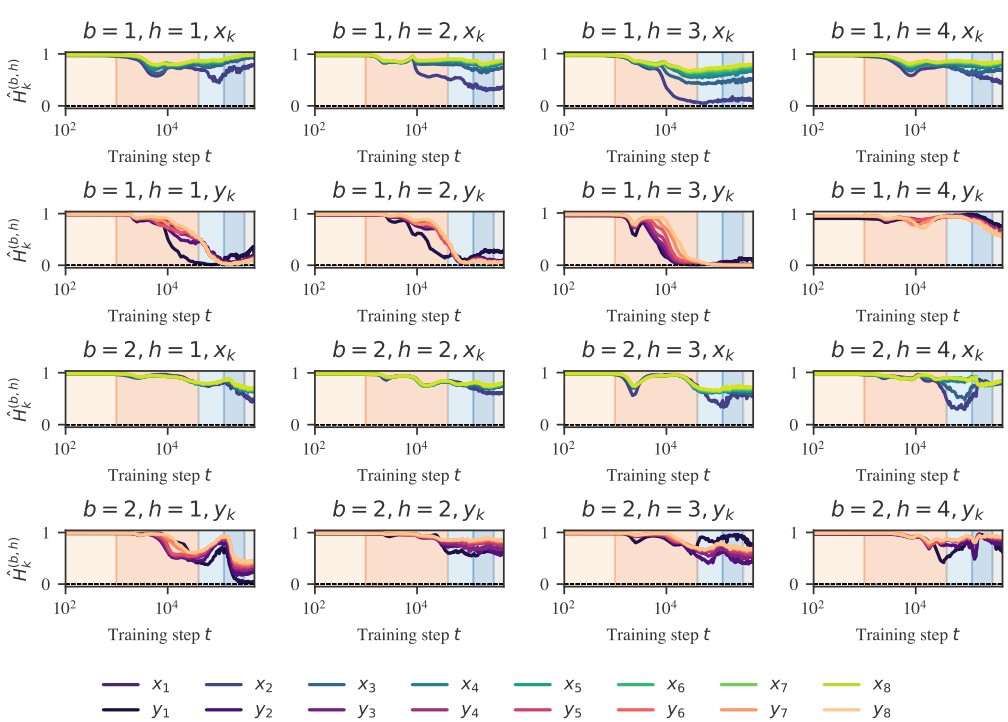

Figure C.8: **Attention hardening** as measured by the normalized attention entropy score (Appendix C.4.5). Block 1 heads 1y/3y and block 2 head 1y harden over training. In combination with the fact that these attention heads become less variable (Figure C.9), this may contribute to a decrease in the LLC (discussed in Appendix C.4.5) The x-components of the attention heads remain much softer over the entire training run.

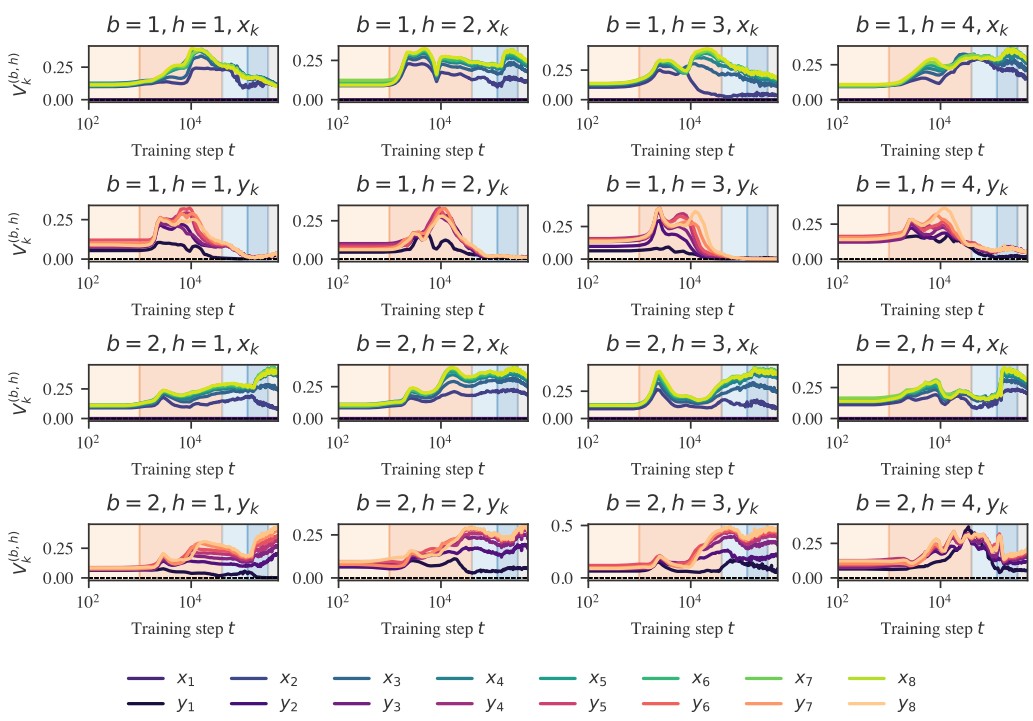

Figure C.9: **Attention variability** over time. The heads that develop hard attention in Figure C.8 (block 1 heads 1y, 3y, and 4y) also become less variable over time.

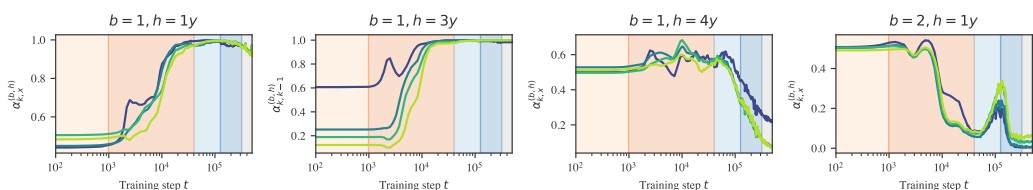

Figure C.10: Collection of attention heads identified by their consistent and recognizable attention patterns. *Left to right*: previous-$x$s head, previous-token head, previous-$y$s head, previous-$y$s head

**$y$-attention score.** Defined analogously $\alpha_{k,x}^{(b,h)} = \sum_{k'=1}^{K} \alpha_{k,2k+1}^{(b,h)}$.

**Classifying attention heads.** Several attention heads are easy to identify by virtue of being both concentrated and consistent. These are depicted in Figure C.10 and include: (B1H3y) previous-token heads (also present in the language model case), (B1H1y) previous-x, and (B1H4x, B2H1y) previous-y heads. Other training runs also include self-attention heads.

**Contributions to degeneracy.** Suppose an attention head $h$ in block $b$ has the following constant attention pattern (after the softmax) $A^{(b,h)} = \sum_i \delta_{l(i)\,i}$. That is, for each token $i$, that attention head attends solely to a single earlier token $l(i) \leq i$ and no others. Restricting to single-head attention (the argument generalizes straightforwardly), the final contribution of this attention head to the residual stream is the following (Phuong and Hutter, 2022):

$$O = W_O \cdot (V \cdot A) \tag{19}$$

where $A \in \mathbb{R}^{\ell_z} \times \mathbb{R}^{\ell_x}$ is the attention pattern, $V \in \mathbb{R}^{d_{\text{out}}} \times \mathbb{R}^{\ell_z}$ is the value matrix, and $W_O \in \mathbb{R}^{d_z} \times \mathbb{R}^{\ell_z}$ is the matrix of residual stream activations, and $V \in \mathbb{R}^{d_{\text{out}}} \times \mathbb{R}^{\ell_z}$ is the value matrix. The result of this operation is subsequently multiplied by the output matrix and then added back into the residual stream. Plugging in the hard and constant attention pattern, writing out the matrix multiplication, and filling in the definition of $A$ we get

$$O_{ij} = \sum_k (W_O)_{ik} V_{kl(j)} \delta_{l(j)j}. \tag{20}$$

For each column in $A$, the hard attention picks out a single element of $V$ at column $l(j)$ for each row $k$. Now suppose that there is a token $l'$ that receives no attention from any position $j$. That is, there exists no $j$ such that $l' = l(j)$. Then, there is a column $l'$ in $V$ which does not contribute to the result of $V \cdot A$, and, in turn, a column $l'$ in $W_O$, which does not contribute to the output of the head. As discussed for the embedding and layer norm, this decrease in effective dimensionality leads to a decrease in the learning coefficient.

Note that this argument does not hold for all hard and constant attention patterns. It holds solely for attention patterns that consistently ignore some earlier token across all positions, such as the previous-$x$ and previous-$y$ heads, but not the self-attention and previous-token heads. As discussed in Appendix C.4.6, it remains unclear what exactly the threshold for "ignoring" a token should be before it contributes to degeneracy and whether any of the heads we examine actually meet this threshold.

### C.4.6 DEGENERACY AND DEVELOPMENT

In the previous subsections, we provide a set of theoretical arguments for how (un)embedding collapse (Appendix C.4.1), layer normalization collapse (Appendix C.4.4), and attention collapse (Appendix C.4.5) can lead to an increase in degeneracy, even while leaving the implemented function unchanged.

The free energy formula tells us that, for two different solutions (sets of weights) with the same loss, the Bayesian posterior will asymptotically prefer the model that has the lower learning coefficient

(i.e., higher degeneracy). This suggests that these different forms of collapse may be driven by a bias towards higher degeneracy, as captured in the free energy formula.

Actually establishing a causal link between increasing degeneracy and structure development is beyond the scope of this paper. For one, the theoretical arguments hinge on the collapse being *complete*, that is, the components that go to zero must become *exactly* zero in the limit, where we take the number of samples to compute the loss to infinity. In practice, we expect there to be some threshold $\epsilon$ below which we can treat weights as effectively zero. Second, even if these explanations are correct, we do not know that they account for all of the empirically observed decrease in the LLC during these stages. There may be other drivers we missed. Finally, establishing a causal link requires theoretical progress in relating the Bayesian learning process to the SGD learning process. The arguments are suggestive, but currently only a source of intuition for how structure and geometry can be related, and a starting point for future research.

## D EXPERIMENTAL DETAILS

### D.1 LANGUAGE MODELS

#### D.1.1 MODEL

The language model architectures we consider are one- and two-layer attention-only transformers. They have a context length of 1024, a residual stream dimension of $d_{model} = 256$, $H = 8$ attention heads per layer, and include layer normalization layers. We also used a learnable Shortformer positional embedding (Press et al., 2021). The resulting models have a total of $d = 3,091,336$ parameters for $L = 1$ and $d = 3,355,016$ parameters for $L = 2$. We used an implementation provided by TransformerLens (Nanda and Bloom, 2022).

#### D.1.2 TOKENIZATION

For tokenization, we used a truncated variant of the GPT-2 tokenizer that cut the original vocabulary of 50,000 tokens down to 5,000 (Eldan and Li, 2023) to reduce the size of the model. We think this may contribute to the prominence of the the plateau at the end of LM1: the frequency of bigram statistics depends on your choice of tokens, and a larger tokenizer leads to bigrams that are individually much less frequent.

#### D.1.3 TRAINING

The models are trained on a single epoch over $50,000$ steps on $\sim$5 billion tokens using a resampled subset of the Pile (Gao et al., 2020; Xie et al., 2023) using a batch size of 100. A snapshot was saved every 10 steps for a total of 5000 checkpoints, though a majority of analysis used checkpoints every 100 steps. The training time was around 6 GPU hours per model on an A100. Additional seeds were trained on v4 TPUs at around 1.5 TPU hours per model.

Training was conducted on the first 10 million lines of the DSIR-filtered Pile (Xie et al., 2023; Gao et al., 2020) but did not exhaust all 10 million lines. The model was subject to weight decay regularization, without the application of dropout. We did not employ a learning rate scheduler throughout the training process.

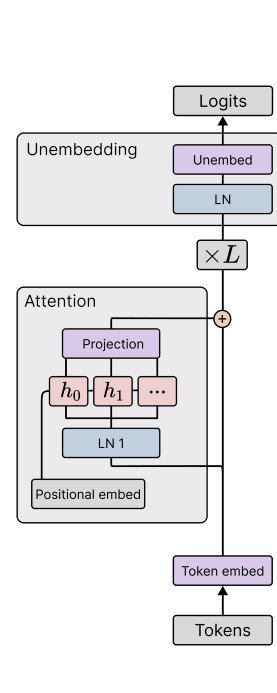

| Component | 1-Layer | 2-Layer |
|---|---|---|
| Token Embedding Weights | 1,280,000 | |
| Positional Embedding Weights | 262,144 | |
| Layer 1 Layer Norm Weights | 256 | |
| Layer 1 Layer Norm Bias | 256 | |
| Layer 1 Attention Query Weights | 65,536 | |
| Layer 1 Attention Key Weights | 65,536 | |
| Layer 1 Attention Value Weights | 65,536 | |
| Layer 1 Attention Output Weights | 65,536 | |
| Layer 1 Attention Query Bias | 256 | |
| Layer 1 Attention Key Bias | 256 | |
| Layer 1 Attention Value Bias | 256 | |
| Layer 1 Attention Output Bias | 256 | |
| Layer 2 Layer Norm Weights | N/A | 256 |
| Layer 2 Layer Norm Bias | N/A | 256 |
| Layer 2 Attention Query Weights | N/A | 65,536 |
| Layer 2 Attention Key Weights | N/A | 65,536 |
| Layer 2 Attention Value Weights | N/A | 65,536 |
| Layer 2 Attention Output Weights | N/A | 65,536 |
| Layer 2 Attention Query Bias | N/A | 256 |
| Layer 2 Attention Key Bias | N/A | 256 |
| Layer 2 Attention Value Bias | N/A | 256 |
| Layer 2 Attention Output Bias | N/A | 256 |
| Final Layer Norm Weights | 256 | |
| Final Layer Norm Bias | 256 | |
| Unembedding Weights | 1,280,000 | |
| Unembedding Bias | 5,000 | |

Figure D.1: **Attention-only transformers** with Shortformer position-infused attention and pre-layer norm. The one-layer model has a total of 3,091,336 trainable parameters, while the two-layer model has 3,355,016.

Table 1: Summary of Hyperparameters and Their Values for Both Language Model

| Hyperparameter | Category | Description/Notes | Value |
|---|---|---|---|
| $n$ | Data | # of training samples | 5,000,000 |
| $T$ | Data | # of training steps | 50,000 |
| $N_{\text{test}}$ | Data | # of test samples | 512 |
| Tokenizer Type | Data | Type of Tokenizer | Truncated GPT-2 Tokenizer |
| $D$ | Data | Vocabulary size | 5,000 |
| $K$ | Data | Context size | 1,024 |
| $L$ | Model | # of layers in the model | 2 |
| $H$ | Model | # of heads per layer | 8 |
| $d_{\text{mlp}}$ | Model | MLP hidden layer size | N/A |
| $d_{\text{embed}}$ | Model | Embedding size | 256 |
| $d_{\text{head}}$ | Model | Head size | 32 |
| seed | Model | Model initialization | 1 |
| m | Training | Batch Size | 100 |
| Optimizer Type | Optimizer | Type of optimizer | AdamW |
| $\eta$ | Optimizer | Learning rate | 0.001 |
| $\lambda_{\text{wd}}$ | Optimizer | Weight Decay | 0.05 |
| $\beta_{1,2}$ | Optimizer | Betas | (0.9, 0.999) |

### D.1.4 LLC ESTIMATION

For LLC estimation, we use SGLD to sample 20 independent chains with 200 steps per chain and 1 sample per step, at a temperature $\beta = 1/\log(m)$, where $m = 100$ is the size of the batch (the maximum size that would fit in memory). For the one-layer model, we used $\epsilon = 0.003, \gamma = 300$, and for the two-layer model we used $\epsilon = 0.001, \gamma = 100$. Estimating the local learning coefficient across all checkpoints took around 200 GPU hours for the two-layer model on a single A100 and around 125 GPU hours for the one-layer model. For additional runs of the two-layer model, we ran fewer chains, bringing the time down to about 2 TPU hours per training run.

We sampled a separate set of 1 million lines (lines 10m-11m) from the DSIR filtered Pile, denoted as $D_{\text{sgld}}$. The first 100,000 lines from this SGLD set (lines 10m-10.1m) were used as a validation set. The sampling of batches for SGLD mirrored the approach taken during the primary training phase. Each SGLD estimation pass was seeded analogously, so, at different checkpoints, the SGLD chains encounter the same selection of batches and injected Gaussian noise.

Table 2: Hyperparameters for Estimating the Local Learning Coefficient for Language Models.

| Hyperparameter | Category | Description/Notes | 1-Layer | 2-Layer |
|:---:|:---:|:---:|:---:|:---:|
| C | Sampler | # of chains | | 20 |
| $T_{\text{SGLD}}$ | Sampler | # of SGLD steps / chain | | 200 |
| $\epsilon$ | SGLD | Step size | 0.003 | 0.001 |
| $\tilde{\gamma} = \epsilon\gamma/2$ | SGLD | Localization strength | 300 | 200 |
| $\tilde{\beta} = \epsilon\beta/2n$ | SGLD | Inverse temperature | | 0.0000217147 |
| $m$ | SGLD | (Default: $\beta^* = \frac{1}{\log n}$) | | 100 |
| | | The size of each SGLD batch | | |
| $\mu$ | Data | Dataset size for gradient minibatches | | 13m |

## D.2 REGRESSION TRANSFORMERS

### D.2.1 MODEL

In the following $L$ refers to the number of layers (blocks) in the Transformer, $H$ is the number of heads in each layer, $D$ is the dimension of inputs $x \in \mathbb{R}^D$ and $K$ is the number of $(x, y)$ pairs provided to the Transformer in-context.

The architecture is a pre-layer-norm decoder-only transformer modeled after NanoGPT (Karpathy, 2022; see also Phuong and Hutter, 2022) with a learnable positional embedding. For the models discussed in the main body, we consider $L = 2, H = 4$ transformers (with $d = 51,717$ parameters), i.e., two transformer blocks with four attention heads each.

### D.2.2 TOKENIZATION

To run contexts $S_K$ through the above model requires an initial encoding or "tokenization step" and final "projection step." The context is encoded as a sequence of "tokens" $T_k$ as follows:

$$T_k = \left( \begin{pmatrix} 0 \\ | \\ x_1 \\ | \end{pmatrix}, \begin{pmatrix} y_1 \\ 0 \\ \vdots \\ 0 \end{pmatrix}, \cdots \begin{pmatrix} 0 \\ | \\ x_k \\ | \end{pmatrix}, \begin{pmatrix} y_k \\ 0 \\ \vdots \\ 0 \end{pmatrix} \right).$$

Through the main text, we write $f_w(S_k)$ for $f_w(T_k)$. Note that this tokenization includes the final $y_k$ token even though this receives no training signal. For this reason, we omit this token from the attention entropy and variability plots (Figures C.8 and C.9).

The transformer outputs a series of tokens of the same shape as $T_k$. To read out the $\hat{y}_k$ predictions, we read out the first component of every other token, i.e.,

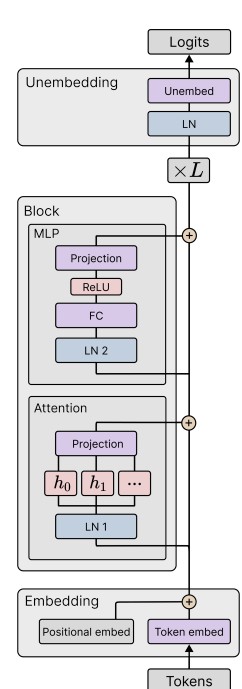

| Component | # of Parameters |
|---|---:|
| Token Embedding Weight | 320 |
| Positional Embedding Weight | 1,024 |
| Layer 1 Layer Norm Weight 2 | 64 |
| Layer 1 Layer Norm Bias 1 | 64 |
| Layer 1 Attention Weights | 12,288 |
| Layer 1 Attention Output Weights | 4,096 |
| Layer 1 Layer Norm Weight 1 | 64 |
| Layer 1 Layer Norm Bias 2 | 64 |
| Layer 1 Feed-Forward MLP Weight | 4,096 |
| Layer 1 Feed-Forward MLP Bias | 64 |
| Layer 1 Feed-Forward Output Weight | 4,096 |
| Layer 1 Feed-Forward Output Bias | 64 |
| Layer 2 Layer Norm Weight 1 | 64 |
| Layer 2 Layer Norm Bias 1 | 64 |
| Layer 2 Attention Weights | 12,288 |
| Layer 2 Attention Output Weights | 4,096 |
| Layer 2 Layer Norm Weight 2 | 64 |
| Layer 2 Layer Norm Bias 2 | 64 |
| Layer 2 Feed-Forward MLP Weight | 4,096 |
| Layer 2 Feed-Forward MLP Bias | 64 |
| Layer 2 Feed-Forward Output Weight | 4,096 |
| Layer 2 Feed-Forward Output Bias | 64 |
| Unembedding Layer Norm Weight 1 | 64 |
| Unembedding Layer Norm Bias 1 | 64 |
| Unembedding Weight 2 | 320 |
| Unembedding Bias 2 | 5 |

Figure D.2: **Transformer parameters in the linear regression setting**. The model has two transformer blocks for a total of $51,717$ trainable parameters.

$$\pi_Y : \mathbb{R}^{2K} \times \mathbb{R}^{D+1} \to \mathbb{R}^K \tag{21}$$

$$\left( \begin{pmatrix} \hat{y}_1 \\ \vdots \end{pmatrix}, \begin{pmatrix} \cdot \\ \vdots \end{pmatrix}, \cdots, \begin{pmatrix} \hat{y}_k \\ \vdots \end{pmatrix} \right) \mapsto (\hat{y}_1, \ldots, y_k). \tag{22}$$

### D.2.3 TRAINING

We train from a single seed for each choice of architecture and optimizer hyperparameters using minibatch stochastic gradient descent. We train without explicit regularization and use the Adam optimizer (Kingma and Ba, 2014). The training runs take 1 to 5 TPU-hours on TPUs provided by Google Research. Models are trained from the same initialization and on the data vectors within each batch (but for different sets of tasks and task orderings).

Models are trained on a single epoch: each of the $T = 500,000$ batches consists of a new set of sequences with batch size 256. For the LLC estimates, we save 190 checkpoints: 100 are linearly spaced over the training run, and the remaining 90 are logarithmically spaced. We perform local learning coefficient estimation and other analyses on these checkpoints.

Table 3: Summary of Hyperparameters and Their Default Values

| Hyperparameter | Category | Description/Notes | Default Values |
|---|---|---|---|
| $n$ | Data | # of training samples | 128,000,000 |
| $B$ | Data | Batch size during training | 256 |
| $T$ | Data | # of training steps | 500k |
| $N_{\text{test}}$ | Data | # of eval samples | 2048 |
| $D$ | Data | Dimensions of linear regression task (Task size) | 4 |
| $K$ | Data | Maximum in-context examples | 8 |
| $\sigma^2$ | Data | Variance of noise in data generation | 0.125 |
| $L$ | Model | # of layers in the model | 2 |
| $H$ | Model | # of attention heads per layer | 4 |
| $d_{\text{mlp}}$ | Model | Size of the hidden layer in MLP | 64 |
| $d_{\text{embed}}$ | Model | Embedding size | 64 |
| seed | Misc | Training run seeds | {0, 1, 2, 3, 4} |
| Optimizer Type | Optimizer | Type of optimizer | Adam |
| $\eta$ | Optimizer | Maximum learning rate | 0.003 |
| $\lambda_{\text{wd}}$ | Optimizer | Weight Decay | 0 |
| $\beta_{1,2}$ | Optimizer | Betas | (0.9, 0.999) |
| Scheduler Type | Scheduler | Type of learning rate scheduler | OneCycleLR |
| Strategy | Scheduler | Strategy for annealing the learning rate | Linear |
| % start | Scheduler | Percentage of the cycle when learning rate is increasing | 0.5 |

### D.2.4 LLC ESTIMATION

For local learning coefficient estimation, we generate a fixed dataset of $2^{20}$ samples. Using SGLD, we sample 10 independent chains with 5,000 steps per chain, of which the first 1,000 are discarded as a burn-in, after which we draw observations once per step, at a temperature $\epsilon\beta/2n = 0.01$, $\epsilon = 0.0003$, and $\epsilon\gamma/2 = 0.01$, over batches of size $m = 1024$. Local learning coefficient estimation takes up to 72 CPU-hours per training run.

Table 4: **LLC estimation hyperparameters**. A summary of the hyperparameters involved in estimating the local learning coefficient and the default values we use.

| Hyperparameter | Category | Description/Notes | Default Values |
|---|---|---|---|
| C | Sampler | # of chains | 10 |
| $T_{\text{SGLD}}$ | Sampler | # of SGLD steps / chain | 5,000 |
| $\epsilon$ | SGLD | Step size | 0.0003 |
| $\tilde{\gamma} = \epsilon\gamma/2$ | SGLD | Localization strength | 0.01 |
| $\tilde{\beta} = \epsilon\beta/2n$ | SGLD | Inverse temperature (Default: $\beta^* = \frac{1}{\log n}$) | 0.01 |
| $m$ | SGLD | The size of each SGLD batch | $2^{10}$ |
| $\mu$ | Data | Dataset size for gradient minibatches | $2^{20}$ |

### D.3 SGLD-BASED LLC ESTIMATION

This section walks through some of the hyperparameter choices and sweeps involved in calibrating LLC estimates. We provide it as a reference for others seeking to adjust LLC estimation to novel settings.

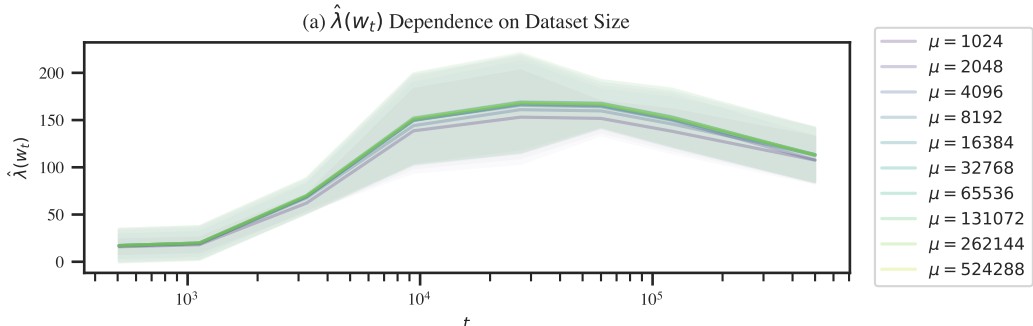

Figure D.3: Past some threshold, the choice of validation set size from which SGLD batches are sampled has little effect on learning coefficient estimates. Estimation hyperparameters are $C = 8, T_{\text{SGLD}} = 2{,}000, m = 2^{10}, \epsilon = 0.0003, \tilde{\gamma} = 0.01, \tilde{\beta} = 0.01$. Loss is evaluated over gradient minibatches at a representative selection of checkpoints. LLCs quickly converge to a constant value as the size increases.

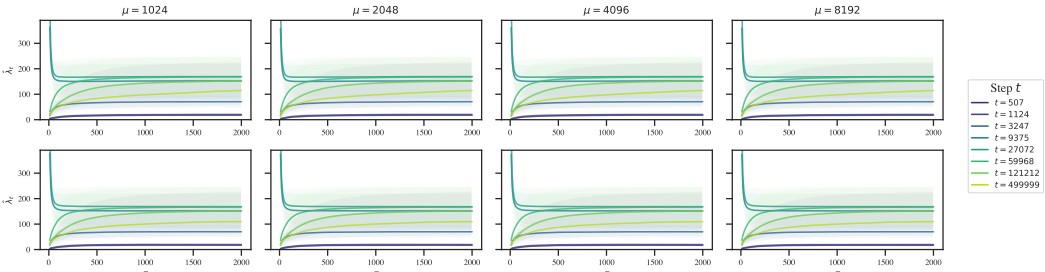

Figure D.4: The size of SGLD minibatches has a negligible effect on LLC estimates (at least among the batch sizes considered). *Top:* Loss is evaluated on the same minibatch as the SGLD gradients. *Bottom:* Loss is evaluated on a newly sampled minibatch from the SGLD gradients (of the same size). Estimation hyperparameters are $C = 8, T_{\text{SGLD}} = 2{,}000, \mu = 2^{20}$.

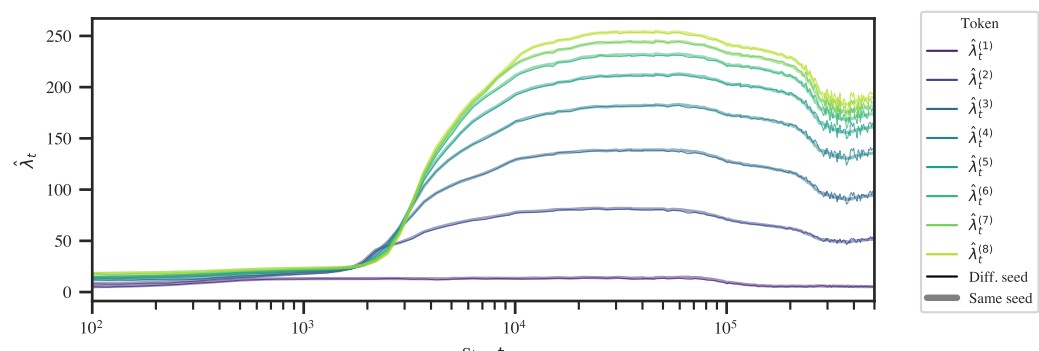

Figure D.5: Consistently seeding SGLD estimates at each checkpoint smooths out the resulting LLC-over-time curve. Except towards the end of training (this is plotted over a log time axis), the difference is barely noticeable. Variable seeds yield a noisier set of estimates.

### D.3.1 VARYING THE TEMPERATURE

In Lau et al. (2024), the inverse temperature $\beta$ is set to a fixed "optimal" value $\beta^* = 1/\log n$, where $n$ is the number of training samples. In practice, we find that it can be advantageous to sample at a higher temperature.

Since $\beta$ always shows up in a product with $n$ (in (8) for the SGLD step and in (3) for the local learning coefficient), we can view the inverse temperature as a multiplier that adjusts the effective size of your dataset. In a Bayesian setting, $\beta = 2$ would mean updating twice on each of the samples in your dataset.

The problem with the default choice of $\beta^*$ is that as we increase $n$ we have to decrease the SGLD step size $\epsilon$ to prevent the update from becoming ill-conditioned, and this eventually causes the gradient term to suppress the noise term. This, in turn, leads to requiring larger batches to suppress the gradient noise and requiring longer chains to sufficiently explore the local posterior (Appendix D.3.3).

Instead of $n\beta = n/\log n$, we perform LLC estimation at $n\beta = m/\log m$, where $m$ is the SGLD batch size.

### D.3.2 SEEDING THE RANDOM NOISE

To smooth out the $\hat{\lambda}_t$ curves, we reset the random seed before LLC estimation run at each checkpoint. This means the sequence of injected Gaussian noise is the same for LLC estimation runs at different checkpoints. Additionally, if the batch size is held constant, the batch schedule will also be constant across different estimation runs. Figure D.5 shows that this does not affect the overall shape of the learning coefficient curves; it simply smooths it out.

### D.3.3 CALIBRATING $\epsilon$, $\beta$, AND $\gamma$

As a rule of thumb, $\epsilon$ should be large enough that the $\hat{\lambda}$ estimate converges within the $T_{\text{SGLD}}$ steps of each chain but not too large that you run into issues with numerical stability and divergent estimates. Subject to this constraint, $\gamma$ should be as small as possible to encourage exploration without enabling the chains to "escape" to nearby better optima, and $\beta$ should be as large as possible (but no greater than $1/\log n$).

To determine the optimal SGLD hyperparameters, we perform a grid sweep over a reparametrization of the SGLD steps in terms of $\tilde{\beta}, \tilde{\gamma}, \varepsilon$:

$$\Delta w_t = \tilde{\beta}\nabla\ell_m^{(\tau)} + \tilde{\gamma}(w^* - w_t) + \mathcal{N}(0, \varepsilon),$$

where $\tilde{\beta} = \varepsilon\beta n/2$, $\tilde{\gamma} = \varepsilon\gamma/2$.

The results of this hyperparameter sweep are illustrated in Figure D.6 for final checkpoints. Separately (not pictured), we check the resulting hyperparameters for a subset of earlier checkpoints. This is

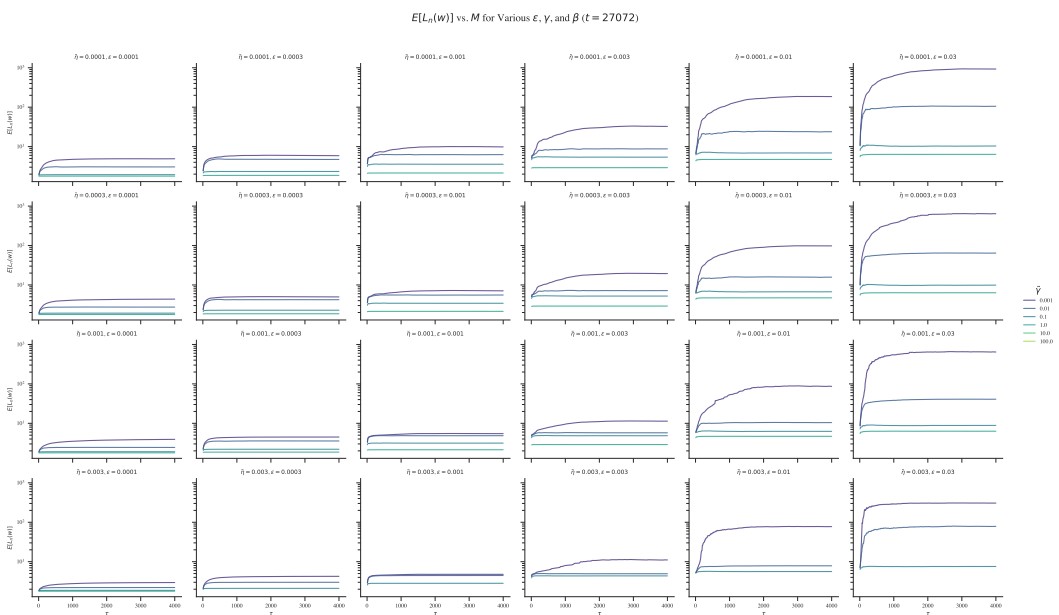

Figure D.6: Results of grid sweep over SGLD hyperparameters for model 0 at $t = 500k$.

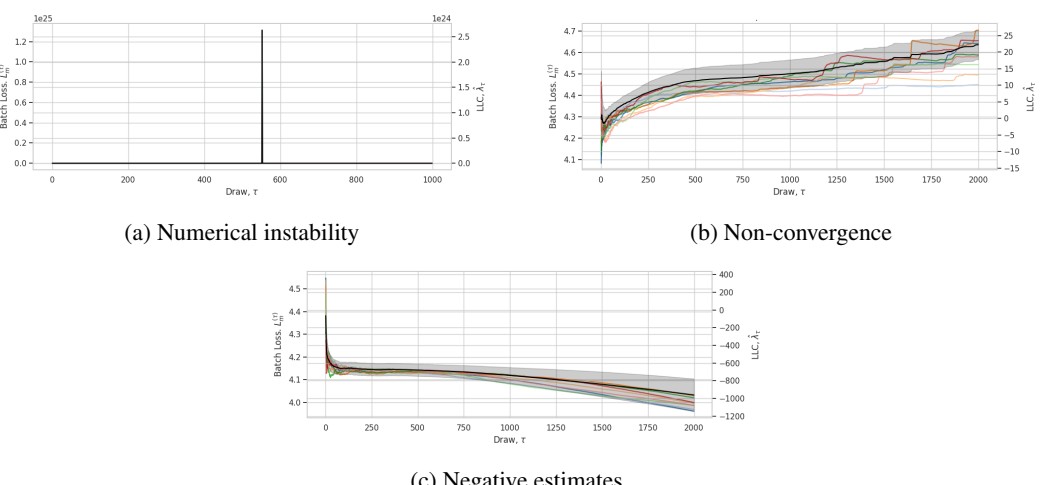

(a) Numerical instability

(b) Non-convergence

(c) Negative estimates

Figure D.7: **Failure modes of SGLD estimation.** *Top left*: the gradient term is too large, leading to issues with numerical instability and exploding $\hat{\lambda}$ estimates. *Top right*: $\epsilon$ is too small, leading to $\hat{\lambda}$ not converging within each chain. *Bottom*: the localization term is too small, which allows the chain to escape to better optima.

needed since, for example, a well-behaved set of hyperparameters at the end of training may lead to failures like divergent estimates (Figure D.7) earlier in training when the geometry is more complex and thus the chains less stable.

### D.3.4   LLC TRACES

As a useful diagnostic when calibrating the local learning coefficient estimates, we propose an online variant for learning coefficient estimation. When overlaid on top of individual-chain LLC traces, this helps reveal common failure modes like divergent estimates, non-converged estimates, and escapes (Figure D.7). These traces display the running estimate of $\hat{\lambda}$ as a function of the number of steps taken in a chain (with the estimate averaged across independent chains).

Define $\hat{\lambda}_\tau(w_0)$, the local learning coefficient at $w_0$ after $\tau$ time-steps for a single SGLD chain as follows (Lau et al., 2024):

$$\hat{\lambda}_\tau(w_0) = n\beta \left( \frac{1}{T} \sum_{t=1}^{T} \ell_n(w_\tau) - \ell_n(w_0) \right).$$

Moving terms around, we get,

$$\hat{\lambda}_\tau(w_0) = \frac{n}{\log n} \left( \frac{1}{\tau} \sum_{\tau'=1}^{\tau} \ell_n(w_{\tau'}) - \ell_n(w_0) \right) \tag{23}$$

$$= n\beta \left( \frac{\tau-1}{\tau} \left( \frac{1}{\tau-1} \sum_{\tau'=1}^{\tau-1} \ell_n(w'_\tau) - \ell_n(w_0) + \ell_n(w_0) \right) + \frac{1}{\tau} \ell_n(w_\tau) - \ell_n(w_0) \right) \tag{24}$$

$$= \frac{\tau-1}{\tau} \hat{\lambda}_{\tau-1}(w_0) + n\beta \left( \frac{1}{\tau} \ell_n(w_\tau) + \left( \frac{\tau-1}{\tau} - 1 \right) \ell_n(w_0) \right) \tag{25}$$

$$= \frac{1}{\tau} \left( (\tau-1)\hat{\lambda}_{\tau-1}(w_0) + n\beta \left( \ell_n(w_\tau) - \ell_n(w_0) \right) \right), \tag{26}$$

where

$$\hat{\lambda}_0(w_0) = 0.$$

This can be easily extended to an online estimate over chains by averaging the update $n\beta \left( \ell_n(w_\tau) - \ell_n(w_0) \right)$ over multiple chains.

## E  ADDITIONAL FIGURES AND CODE

For additional figures and code, see the anonymized repository located at https://anonymous.4open.science/r/icl-0C47.

