# OpenReview forum: "Stagewise Development in Transformers and the Geometry of the Loss Landscape"
_ICLR.cc/2025/Conference — Submitted to ICLR 2025_

### Official Review · Reviewer_3gci · 2024-11-03

**Soundness:** 4
**Presentation:** 3
**Contribution:** 3
**Rating:** 6
**Confidence:** 2

**Summary:**

This paper leverages the LLC as metrics to establish the developmental stages for language modeling of transformer and in-context linear regression. These stages reveal the developmental changes of models in language modeling.

**Strengths:**

1.	The authors analysis the behavioral and structural changes of each stages to confirm the meaning of LLC, and each stage actually has the valuable behavioral changes to reveal the learning process in the language modeling.
2.	The paper uses the realistic data to train analysis the language modeling of transformer.
3.	The experiment is adequate and reasonable and the paper is well written.

**Weaknesses:**

1.	The used transformers are one and two layer attention-only. Whether the same behavioral and structural changes can be seen in more larger transformer. The scaling may affect the loss landscape.

**Questions:**

The same as above.

---

> ### Author Response · Authors · 2024-11-18
>
> Thank you for your review and for recommending that the paper should be accepted. We appreciate your positive feedback and you raising your concern about generalizability.
>
> **On generalizability to larger models:** We appreciate your concern regarding the scalability of our methodology. Please see our top-level comment (on the scale of experiments) for a detailed response. Importantly, we agree that the specific stages that we identify may not generalize to larger models. In fact we do not claim that these stages will generalize to other models or with other hyperparameters. The thing that we claim is that we show evidence for the existence of a fundamental link between loss landscape geometry and neural network development. If this link truly exists, this is the thing that we think will generalize to larger models, which would allow future more sophisticated geometry-based methodologies to better understand development in larger models.
>
> **Thanks again\!** Given the new information presented in our top-level rebuttal clarifying the aims of our work, we wanted to ask if this addresses your concern about generalizability and whether you would consider strengthening your recommendation to accept the paper by improving your score? Either way, we welcome further discussion.

---

> ### Author Response · Authors · 2024-11-24
> **Follow-up: Have our clarifications strengthened your impression of the paper?**
>
> Dear Reviewer 3gci,
>
> Thank you once again for your encouraging support of our paper. While there is still time in the discussion period, we wanted to check in with you.
>
> We have taken steps to clarify the contributions and broader implications of our work. In particular we noted that our primary claim is not that the stage-detection methodology will necessarily generalize to different or larger models, but that we have contributed evidence of a deep link between loss landscape geometry and neural network development that has the potential to generalize (among other implications).
>
> We wanted to ask whether you have considered the revised paper's new introduction and discussion sections, where we have made a clearer and more explicit case for the contribution of the paper than in the original submission. Has this new perspective at all strengthened your impression of the paper? If so, please consider raising your score.
>
> Of course, if you have any remaining concerns or follow-up questions prompted by our revisions or the discussion with the other reviewers, we welcome further discussion.

---

### Official Review · Reviewer_E5MY · 2024-11-04

**Soundness:** 2
**Presentation:** 3
**Contribution:** 2
**Rating:** 5
**Confidence:** 4

**Summary:**

This paper introduces the Local Learning Coefficient (LLC) as a novel approach to explain phase transitions in transformer training dynamics. LLC is calculated by measuring the expected difference between the posterior and local optimal solutions. The authors establish connections between LLC and the geometry of model likelihood, offering a new perspective on transformer learning behavior.

**Strengths:**

- Introduces a new metric (LLC) for analyzing transformer training dynamics and phase transitions, providing an alternative to traditional per-token loss measures

- Presents comprehensive experimental analysis with thorough ablation studies

- Provides detailed methodology and experimental setup documentation

**Weaknesses:**

- The relationship between loss changes and LLC lacks clear description. While Table 1 presents changes in both ICL loss and LLC, it
 does not establish a definitive connection to phase transitions. In Figure 1 the dynamics of ICL loss and LLC do not match or describe the phrase transitions.  The paper can benefit from quantitative correlation metrics between ICL and LLC to strengthen the qualitative observations.

- Despite extensive analysis, the paper primarily reinforces existing findings from Olsson et al. (2022) rather than presenting novel insights into transformer learning dynamics

**Questions:**

1. What are the specific advantages of LLC over per-token loss that justify its adoption as a preferred metric for analyzing transformer training dynamics?

2. The LLC computation relies on finding $w*$ (local minimum) through stochastic Langevin dynamics. How can LLC be reliably estimated with respect to weights $w_t$ during the actual training process?

---

> ### Author Response · Authors · 2024-11-18
>
> Thank you for your review and for recognizing several strengths of our analysis approach. We also appreciate you sharing your concerns and questions about the work to which we offer the following comments.
>
> **On the relation between loss changes and LLC:** We are not making a claim to a definitive connection between our developmental stages and phase transitions in the sense of statistical physics in this paper, although we note that prior work \[1\] does explore this in smaller models. Could you please clarify what you mean by the “ICL loss and LLC do not match”? The only implied match in this paper is between critical points of the LLC curve and the stage boundaries.
>
> **On novel insights into transformer training dynamics:** We respectfully disagree with the claim that our work “primarily reinforces existing findings from Olsson et al. (2022) rather than presenting novel insights into transformer learning dynamics”.
>
> * **We do more than reinforce the results of Olsson et al. (2022), we extend them.** It’s true that we replicate the emergence of induction heads in two-layer attention-only transformer language models observed by Olsson et al. (2022). However, we also show that before induction heads form, our transformer adopts simpler strategies, following a progression akin to that found by Olsson et al. (2022) for fully-developed models with increasing depth. To our knowledge this is a novel contribution to the literature on transformer learning dynamics.
> * **Our contributions include more than just our observations about the specific dynamics of these transformers.** Please see the discussion in the top-level comment on primary takeaways, where we enumerate several other novel implications of the extensive analysis that we contribute.
>
> We appreciate your comment because we realized we could have made the significance of our contributions and the relationship to prior work much clearer in the paper. We have elaborated on these points in the revised discussion section. Please take a look at the revised discussion and let us know if you have any further concerns.
>
> **On the LLC vs. per-token loss:** Great question. The primary advantage of the LLC over the per-token loss is that the LLC reveals additional information that is hidden from the per-token loss, offering a complementary perspective on model development. We can see this empirically in our results and it is also consistent with singular learning theory.
>
> * Our results in in-context linear regression demonstrate this empirically: the in-distribution input-output mapping largely stabilizes after LR2. This is apparent in the per-token loss as well as per-token prediction norm (Figure 6a (was 4a), top). However, LLC continues to reveal significant developmental changes in stages LR3–LR5, including phenomena like layer normalization collapse that are invisible to purely behavioral metrics.
> * In general, we expect (per-token) loss and the LLC to be highly complementary metrics. The LLC is grounded in singular learning theory through the free energy formula (Section 4 (was 3.1)), which provides a principled mathematical basis for using it to detect developmental stages and transitions in model complexity. Singular learning theory suggests that the LLC is, after the loss, the second most important observable for neural networks. Thus the two metrics would best be studied together.
>
> **On estimating LLC away from local minima:** Another great question. We have addressed this in the top-level comment. Does the response there address your question?
>
> **Thanks again\!** We once again appreciate your feedback and questions. We are not 100% sure that we have addressed the relationship between loss changes and LLC, but look forward to further discussion. We feel that through the above response and our top-level comment and revisions we have addressed the claim that we have not offered novel results, and we are wondering if this is enough for you to reconsider your recommendation to reject the paper. Of course, if you have any other follow-up questions we are happy to elaborate.
>
> \[1\] Chen et al. (2023); as cited in manuscript.

---

> ### Author Response · Authors · 2024-11-24
> **Follow-up: Please let us know whether we have addressed your concerns**
>
> Dear Reviewer E5MY,
>
> Thank you once again for your review. While there are still a few days left of the discussion period, we wanted to quickly follow-up regarding your review. We are still hoping to discuss further with you about your listed concerns and questions:
>
> 1. We are still seeking clarification of your first stated concern, regarding quantifying the relationship between ICL, loss, and LLC. We have clarified that we make no claims about establishing "a definitive connection to phase transitions".
> 2. Regarding our contributions relative to prior work, we feel we have addressed this concern. We have clarified in our individual response, in our top-level comment, and in the new introduction in our revised paper that we draw several novel takeaways from our extensive analysis.
> 3. In response to your question about the advantages of LLC over per-token loss, we have clarified how we see the two metrics as complementary, which is a perspective supported by our empirical results and by the framework of singular learning theory.
> 4. In response to your question on the use of LLC estimation for points away from local minima, we have clarified that our methodology is empirically supported by prior work in smaller models, which we see as sufficient for us to draw the conclusions we draw (and while a full theoretical justification for this approach is currently lacking, we have prominently flagged this as a limitation in the revised paper).
>
> Please let us know whether our responses have been sufficient to allay your concerns about the paper. In the case of point (1), we would greatly value clarification so that we can better understand and then address this concern. If we have indeed addressed your concerns, we once again invite you to consider increasing your rating. Otherwise, if you have remaining or new questions or concerns, we invite you to share them so that we can make use of the remainder of the discussion period.
>
> Thank you again for the time and attention you have paid to our work.

---

### Official Review · Reviewer_sxSg · 2024-11-04

**Soundness:** 2
**Presentation:** 2
**Contribution:** 3
**Rating:** 5
**Confidence:** 3

**Summary:**

This paper employs "Local Learning Coefficient" (LLC), a recently proposed metric, to measure the geometry of loss landscape during training transformers.

By conducting experiments on two-layer attention-only transformers on some simple tasks, the authors find that the training could be divided into several discrete development stages according the LLC features.  And then different behavioral and structural features are found among different stages.

This work could bring meanings to reveal the learning mechanistic (and decrete development) for transformers.

**Strengths:**

1. This work finds there are some discrete development stages for transformers via analyzing the geometry of loss landscape. It could bring insights for related works about mechenistic interpretability to transformers.

2. I like the analyses in Stage validation section, which tries to connect LCC trend and some visible important features, though I also have some questions about this section.

On the whole it is an interesting work and I expect to have more discussion in rebuttal period. I am open to improve the score if my following concerns are well addressed.

**Weaknesses:**

1. I am concerned about the theory adopted in this paper, LLC. It seems that the theory of LLC is not widely accepted in the research community and may subject to potential pitfalls and limitations. The lack of peer-reviewed studies on LLC means that we have a limited understanding of its applicability, reliability, and overall validity.

2. In Section 4.3, the authors state that First-layer previous-token heads start to form, and the evidence is that Figure 2(b) top. However, I think it is more like a confuse cause and effect. After the authors discover that two specific heads start to have high previous-token score in LM3 stage, the previous-token heads, 1:2 and 1:5, are then indicated. Whilest, In LM2 stage, there are already many heads that get high scores, why aren't they the previous-token heads? Furthermore, LM3 seems less meaningful compared to other stages.

3. I think this paper might neglect the writing of some specific experimental implementation methods, which needs to be clarified more.
- LCC is based on the measure of loss. How do you measure loss? Specifically, what dataset or distribution (and how many samples) do you use to measure loss? Would different dataset lead to totally different results? (that is, different loss landscapes for different validation datasets).

[1] Lau, Edmund, Daniel Murfet, and Susan Wei. "Quantifying degeneracy in singular models via the learning coefficient." arXiv preprint arXiv:2308.12108 (2023).

**Questions:**

1. In Figure 1, would the training process contain more stages (i.e. d(LCC)/dt = 0 points) if you lengthen the training? If there are more stages, what is the corresponding features (behavior and structure) and why do you think the first 5 stages are the most important?

2. What is the mathematical basis for d(LCC)/dt = 0 points? Why are these points critical and able to become the boundary of development stages (from mathematical perspective)? Otherwise, is this a target-guided results (that is, we have discrete stages first and we dig the math features of the boundary)?

3. I notice that this paper did not employ a learning rate scheduler throughout the training process. Though to some degree I acknowledge the setup for controlling variables, learning rate is a very important factor to determine the loss landscape. Many previous works point out that lower learning rate helps models to converge to a local minimum, and decayed learning rate can help models generalize better due to more flattened local minimum. What do you think about the impact of learning rate schedule on your work?

---

> ### Author Response · Authors · 2024-11-18
> **Response (part 1)**
>
> Thank you for your detailed review and for sharing your concerns and questions about the paper. Thanks especially for acknowledging the potential of this research to complement work on mechanistic interpretability. We are looking forward to further discussion.
>
> **On theoretical foundations:** We appreciate you raising this concern. We think we have a strong response.
>
> * First, we claim that our methodology is deeply principled. The core theoretical foundation of our approach is based on singular learning theory, which is an established body of work published in mathematics, statistics, and machine learning communities for over two decades. It’s true that the particular tool that we use for quantifying degeneracy, Lau et al.’s SGLD-based LLC estimator, is a recent idea that, to our knowledge, has not yet been published in a peer-reviewed deep learning venue.
> * Second, in light of the clarification in our top-level comment about the primary takeaway of our work, we contend that in order to establish our main claim (that our results point to a link between geometry and development), it is not necessary to have a perfect LLC estimation methodology. We contend that we have contributed evidence supporting this link to a sufficient degree that warrants publication, so that future work (including work on improved LLC estimation approaches and more extensive replication of our findings in a broader class of settings) can continue to investigate this link and its implications for better understanding neural network development.
>
> In recognition of the fact that the LLC is not a widely-used metric within the deep learning community, we have rewritten the methodology section in our latest revision. This includes offering a more detailed and accessible introduction to the LLC, to Lau et al.’s SGLD-based LLC estimation procedure, and to the singular learning theory ideas motivating the other parts of our stage-identification methodology (e.g. using critical points of the estimated LLC curve to mark stage boundaries). We hope this makes our paper more self-contained.
>
> We hope that this explains why we feel confident using Lau et al.’s SGLD-based LLC estimation procedure as one part of our methodology even though it has not been peer-reviewed. Have we allayed your concern on the matter?
>
> **On figure 4b (was 2b):** Your main question as we understand it is, why are the previous token heads identified as such when there are other heads that increase in their previous token attention score in LM2? Our identification of heads 1:2 and 1:5 as previous token heads is due to the combination of the previous token score and the composition score of those heads with the induction heads, which can be seen in Figure B.5. The other attention heads that increase in previous token attention score in LM2 do not have a high composition score with the induction heads. So, they do not form part of the induction circuit. (As for why other heads have their previous token score increase in LM2, these heads are responsible for predicting the $n$-grams that are learned in LM2, which naturally requires an attention head to attend to recent token positions.)
>
> We appreciate you raising this concern and giving us the opportunity to improve the section by clarifying these points explicitly. We have attempted to do so in the revision, please let us know what you think.
>
> **On data distribution for computing loss:** We appreciate your feedback. Regarding your questions about the validation data used for LLC estimation:
>
> * We have documented some of these details in the appendices, namely in section D.1.4 and table 2 (was 3\) for language modeling, and section D.2.4 and table 3 (was 4\) for in-context linear regression, which describe the data distributions and numbers of samples used in SGLD-based LLC estimation. We realized we didn’t link these appendices clearly from the main text—we have corrected this in the revision.
> * As described in these appendices for SGLD we use an independently sampled data set (independent from the training data set; a different subset of The Pile for language modeling, and a fresh sample of synthetic data from the same distribution for in-context linear regression). In the process of SGLD we compute the loss based on minibatches drawn from these data sets.
> * Of course, there will be small fluctuations in the loss landscape if we generate different data samples or different batches, but since we consider large samples from the same underlying distribution, we don’t think this kind of resampling would cause the SGLD loss landscape to change enough to create a meaningful difference in our estimates.
>
> Is this information satisfactory, or is there anything else we can clarify? In our revision we have more clearly indicated the location of this information in the appendices at the appropriate point in the main text. Does that resolve your concern about the paper not having enough clarity on important experimental details?

---

> ### Author Response · Authors · 2024-11-18
> **Response (part 2)**
>
> **On longer training:** We believe that there are no additional stages to be found by lengthening training, at least with the current hyperparameters. Notice the log scale on the x-axis for e.g. Fig 1, so that in the language model setting the final stage is from 17k-50k steps, and in the linear regression setting from 320k-500k. So the model spends in both cases a significant fraction of the training time in the final stage, and we see no evidence that longer training would reveal interesting new development.
>
> **On the mathematical basis for d(LLC)/dt \= 0:** Great question. The link between SGD training and the way the Bayesian posterior changes in the singular learning process remains unclear, so it is not possible to give a precise justification for looking for points where d(LLC)/dt \= 0 at the present time. However the motivation is as follows: **i**n smaller models (such as \[1\]) the developmental stages correspond to the learning trajectory passing from the neighborhood of one critical point of the population loss to another, and each of these critical points has an associated LLC value. If the trajectory is experiencing “critical slowing down” near one of these critical points, the LLC estimate should stabilize near this value and this is exactly what was observed in \[1\]. In larger models the picture is more complex and not fully understood, and a simple picture of the learning trajectory passing near critical points is unlikely to be correct. Nonetheless our overall expectation is still that at some points in training the LLC estimate should stabilize when important structures are “nearing completion” and this motivates the d(LLC)/dt \= 0 criterion.
>
> We appreciate your question because this reflects the fact that we did not spell out this connection in the submitted version of the paper. In our latest revision, our updated methodology section (now Section 4\) spells out this motivation.
>
> **On learning rate schedulers:** We agree that a dynamic learning rate could influence the developmental stages and think this is an important direction for future work. Our speculation would be that this would mostly affect the later stages in training, on the theory that these are about learning “finer” patterns in the data and the discovery of these patterns is what a decaying learning rate is enabling. So a reasonable expectation would be that with a decaying learning rate the early stages would look the same, but (returning to your earlier question) there might be additional structure to be uncovered in the final stage which is currently invisible.
>
> **Thanks again:** We hope we have been able to address most of your questions and concerns about the paper. If so, we invite you to reconsider your recommendation to reject the paper. If you still have concerns or if you have any follow-up questions, please do let us know. Looking forward to further discussion.
>
> \[1\] Chen et al. (2023); as cited in manuscript.

---

> ### Author Response · Authors · 2024-11-24
> **Follow-up: Have you had a chance to consider our response?**
>
> Dear Reviewer sxSg,
>
> Thank you again for your engagement with our paper and for your openness to further discussion. While there are still a few days left of the discussion period, we wanted to quickly follow-up on this.
>
> We have attempted to address most of your stated concerns about the paper:
>
> 1. Regarding the concern that the theory is not based on widely-accepted prior work, we have pointed out that the LLC estimation techniques we use are new but, like the rest of our methodology, are soundly based on a firm foundation of published work in singular learning theory. We have also attempted to clarify the presentation of our methodology and how it is derived from this prior work in sections 3 and 4 of our revised paper, which we invite you review.
> 2. Regarding the concern about the classification of previous token heads in figure 2b (now 4b), we have clarified that we identified these heads based on their role in the induction circuit and not only on their previous token score.
> 3. Regarding the concern about missing experimental details for measuring the loss during LLC estimation, we have pointed out these details in the appendix and we have included more prominent links to these details in the revised main text at the appropriate place.
>
> We invite you to share any follow-up questions, or let us know if the above responses have not sufficiently addressed your concerns, or if you have any other remaining concerns. Otherwise, we invite you to reconsider your rating. We are very much looking forward to a fruitful discussion over the remainder of the discussion period.

---

> > ### Comment · Reviewer_sxSg · 2024-11-25
> > **Follow-up questions**
> >
> > Thanks for your reponse. Here are some of my follow-up questions:
> >
> > **On theoretical foundations:** This response addresses part of my concern to some degree. **I still recommand other reviewers (maybe including AC) who are more familiar with this field to pay more attention to this issue.**
> >
> > **On data distribution for computing loss:** Thanks for the clarification. It seems that one of my questions was not well answered. Would different dataset lead to totally different results? What if you use another dataset as validation set other than The Pile? For example, Python code corpus. From my perspective, the geometry might change a lot because the validation loss values change a lot.
> >
> > **On longer training:** It is still concerning if there would be more stages. Especially, the conclusion (there are only 5 stages) is based on some toy/simple experimental settings (e.g., no learning rate schedule as I mentioned before). In a more complex setting, such finding could not totally fit any more.

---

> > > ### Author Response · Authors · 2024-11-25
> > >
> > > Thanks for the questions.
> > >
> > > **On data distribution for computing loss**. Regarding the question of whether a different dataset would lead to different results. This is correct: the loss landscape and its geometry is a function not only of the architecture of the model but of the data distribution, and if we use a different data distribution to estimate the LLC we will be probing the geometry of a different loss landscape. In the present paper we measure the LLC only using the Pile, because this is the pre-training distribution and when we are studying the development of the model over training, it is the geometry associated to the pre-training distribution that is a priori the most relevant. We view the fact that this LLC curve has critical points that reflect the developmental stages of training to be some evidence that this geometry is indeed relevant.
> > >
> > > However, measuring the LLC using other data distributions (probing "other loss landscapes at the same parameter") is theoretically meaningful and we do think this is an interesting direction to pursue. As you suggest, measuring the LLC with respect to a corpus of code should in particular offer information about the amount of information in the model about code (which might develop differently to the overall information about the broader distribution of the Pile). Concurrent work has looked at this, but before sharing a direct link here we will consult with the program chair due to a question about de-anonymization.
> > >
> > > Just to be clear, the fact that the LLC estimate will change with the data distribution chosen doesn't undermine the value of studying it for the pre-training distribution; this is the distribution that governs the training process, and the corresponding LLC estimate is the right complexity measure for studying training.
> > >
> > > **On longer training**. Our claim is that the methodology of using LLC estimation reveals that transformers learn in stages, a fact not immediately visible in the loss and not previously well-studied. Although we carefully document five stages in these particular models in order to establish this fact, it is not important to us that there are *exactly* five stages (and indeed we do not claim that larger transformers will have these precise five stages, either). The generalisable claim we want to make is that there is a link between developmental stages and loss landscape geometry, and that our methodology provides a tool for finding those stages. If future work digs more closely into the final training stage and finds evidence for distinct substages within it, or trains for longer and finds additional stages, this would be interesting and welcome. In our opinion this does not detract from the contribution of the present paper.
> > >
> > > As mentioned above, as far as we can tell given our computational budget, we see no evidence that longer training would reveal interesting new developments, but we can of course not rule this out. On reflection it seems this is not well communicated in the current draft and we will revise the language.

---

### Official Review · Reviewer_PMP3 · 2024-11-04

**Soundness:** 3
**Presentation:** 2
**Contribution:** 3
**Rating:** 6
**Confidence:** 2

**Summary:**

This paper investigates the emergence in stagewise development of internal structures in the training of transformer models, specifically for tasks like language modeling and in-context linear regression. The authors utilize the Local Learning Coefficient (LLC) to measure geometric degeneracy in the loss landscape and identify distinct training stages of the model’s behavior and internal structures. Beyond general loss decreases in model training, the authors discover several stages in which the geometry becomes more degenerate linking to a phenomenon named “layer normalization collapse”. These findings provide valuable insights into the complex processes of transformer training and underscore the importance of loss landscape geometry in understanding model development.

**Strengths:**

1. The paper introduces a novel method Local Learning Coefficient (LLC) to identify the stage boundaries
2. The evaluation methods proposed in this article have been verified for two different types of tasks providing an enhanced understanding of transformer model training.
3. Bring a new insight to the model stability and robustness by showing a phenomenon of “layer normalization collapse”.

**Weaknesses:**

1. This method has certain limitations in model selection. The language model only has 3 million parameters. Perhaps these methods cannot be directly generalized to large models such as GPT and BERT. Therefore, the universality of this model needs to be verified.

2. The LLC method has certain limitations. Because it's estimation will be affected by the training parameters. When the parameters are not the local minimum of the loss, the estimation of LLC might have some bias. Though the article attempts to estimate through the SGLD method, since LLC is sensitive to hyperparameter selection, this instability will affect the credibility of experimental results.

3. The author proposed the concept of "layer normalization collapse", but lacked some more in-depth discussions such as the causes and some quantitative analysis. These analyzes will add to the value of this study.

**Questions:**

1. Can this method be extended to a larger range of tasks such as image processing? Can similar patterns also appear?

2. Do you have theoretical reasons to believe your method would or would not scale to larger models?

3. Could you provide metrics on the rate or extent of collapse across different model sizes or training regimes.

For other questions, please refer to weakness.

---

> ### Author Response · Authors · 2024-11-18
>
> Thank you for your review and for sharing your concerns and questions about our work. We would like to respond to each of your concerns and questions in detail.
>
> **On limitations of model selection:** Please see our top-level comment (on the scale of experiments). Importantly, we agree that our stage identification methodology may not transfer to larger models, and we do not claim that it will. The models are merely case studies, chosen to have non-trivial scale while still being within the realm of interpretability (which is required for our validation step). We claim their scale is sufficient for our results to demonstrate meaningful evidence for the existence of a fundamental link between geometry and development in deep learning. As you point out in your review, we also demonstrate this link in two different settings. Do you agree?
>
> Regarding your question (Q2), in theory our method should scale to larger models, with the caveats in the top-level comment. In more detail, LLC estimation works in principle for any architecture as long as we can run SGLD. The free energy formula underlying our stage identification methodology holds equally for larger models.
>
> As discussed in the top-level comment, the primary barrier to investigating larger models is not to stage identification, but only to validation: interpreting the developmental stages might be more challenging due to the increased model complexity.
>
> **On LLC estimation away from minima:** We agree this is a limitation, but we do not think it invalidates our approach. Please see the top-level comment for details, and let us know what you think.
>
> **On sensitivity of LLC estimation:** We adhered to guidelines established in prior work on LLC estimation for selecting SGLD hyperparameters while minimizing bias in LLC estimates. We detail our procedure in appendix D.3. Does this approach address your concern?
>
> **On layer normalization collapse:** We appreciate your interest in this phenomenon. You mentioned that you saw it as a strength that it can bring new insights to model stability and robustness. We wanted to clarify that we don’t see layer normalization collapse as necessarily having implications for model stability or robustness. Rather, we see it as a possible mechanism behind the phenomenon that in some stages, the LLC decreases (please see the top-level comment, extra takeaway 3, for further clarification).
>
> As you also note under weaknesses, we could have done more to clarify the implications of this phenomenon, and to provide an in-depth analysis. In our latest revision, we have included additional discussion on how layer normalization collapse (along with other observed changes in structure in this model) connects to an increase in degeneracy in appendix C.4. In C.4.6 we discuss how these degeneracy decreases can be consistent with the free energy formula, but the precise interpretation remains an open theoretical problem.
>
> Regarding your question (Q3) about the rate and extent of LN collapse:
>
> * **Extent:** The LN collapse is near-complete for the unembedding (see insets in Figure C.7): more than 62/64=97% of the unembedding LN weights are less than 0.1 in magnitude by the end of training. The next-largest collapse occurs in the block 1 pre-attention LN weights, where about half of the weights collapse. In the remaining LN layers, the weights do visually appear to collapse (main plots in Figure C.7), but the weights do not reach as small a norm.
> * **Rate:** The collapse occurs over the course of stages LR3 and LR4 at the same time for all of these different layers. Individual weights can collapse in as few as \~10k steps and in other cases closer to \~100k steps (main plots in Figure C.7).
>
> In other seeds we observe the same extent and rate of collapse. We have not explored the collapse in other training regimes, though this is an interesting direction for future work, especially as it relates to the link between geometry and development.
>
> **On stages in image models:** Thank you for this intriguing question (Q1). We have not yet studied links between geometry and development in image models. However, we see this as a promising avenue for future work, as the more disparate settings in which we see evidence for links between geometry and development, the more support this could provide for the fundamental nature of this link.
>
> **Thanks again.** Has the above reply and the clarification of the aims of our work in the top-level comment resolved most of your questions and concerns about the paper? If so, please consider raising your score. Also, please let us know if you have any follow-up questions, we are looking forward to further discussion.

---

> ### Author Response · Authors · 2024-11-24
> **Follow-up: Have we addressed your concerns?**
>
> Dear Reviewer PMP3,
>
> While there are still a few days left of the discussion period, we wanted to quickly follow-up regarding your review and our response. We feel we have addressed most of your stated concerns about the paper:
>
> 1. Regarding the generalization to larger models, we have clarified that we do not claim generalization to larger models, and our top-level comment and revised paper clarifies what we do see as the takeaways from our work, centering around the idea that our experiments in two settings provide evidence of a fundamental link between geometry and development.
> 2. Regarding the limitations of local learning coefficient estimation, we have clarified how we went about tuning the estimates to ensure our results are robust.
> 3. Regarding the lack of details on layer normalization collapse, we have clarified that we see this as a potential mechanism for the phenomenon of model simplification and we have added additional details in our response and in our revised paper.
>
> Could you please confirm whether the above has addressed your concerns? If so, we invite you once more to reconsider your rating and share any follow-up questions prompted by our response or the discussion with other reviewers. Thank you again for your consideration, we look forward to further discussion.

---

> > ### Comment · Reviewer_PMP3 · 2024-11-24
> > **Thank you for your explanation**
> >
> > Thank you for your clear and complete explanation. I am willing to improve my score because your answer solved most of my doubts.

---

### Author Response · Authors · 2024-11-18

Thank you all for the time and attention you have contributed to our submission. We have replied to each of you in detail. We also wanted to expand on a few common themes here.

**Primary takeaways of the work:** Several reviewers summarized our work as primarily contributing a novel LLC-based stage-identification methodology. While this methodology plays a central role in our work, we don’t see the methodology itself as our most important contribution. Rather, the fact that this methodology uncovers meaningful developments suggests that **there is a fundamental link between loss landscape geometry and neural network development**—this link is what we see as the most important takeaway.

Beyond the geometry–development link, we see three additional takeaways, as noted by some reviewers:

1. **Case studies in transformer development:** our analyses of the specific models we train under specific training conditions contributes a small amount to a growing literature on the phenomenology of the development of in-context learning in transformers. For example, we see our language models move through a progression of algorithms previously known to be implemented by fully-trained models of increasing numbers of layers.
2. **Developmental interpretability:** our geometry-based stage-identification methodology has shown it is able to detect bulk-level changes in model behavior and structure, without a priori mechanistic understanding. Thus it provides a setting-agnostic alternative to setting-specific progress measures, and could in principle lead to future techniques for detecting important changes in models as they are trained.
3. **Sometimes the models develop into a simpler form:** We see several examples of stages in which the LLC decreases, indicating the model undergoing a spontaneous simplification. While we have not been able to uncover the mechanistic nature of the simplification in this case except for noting the potential connection to observable shifts in certain weights, this phenomenon is noteworthy because it is currently not accounted for by existing theories of neural network development.

In the following comment we explicitly list and discuss these takeaways in the introduction and discussion of our latest revision. We hope they can be taken into consideration when evaluating the paper, and we welcome further questions.

---

> ### Author Response · Authors · 2024-11-18
>
> **On the scale of experiments:** Reviewers PMP3 and 3gci raised concerns that while our results hold for two-layer transformers, our methodology might not generalize to LLMs. We want to clarify our choice of models and the ways in which we do and do not expect our results to generalize.
>
> As explained, our primary claim is that our results point to a link between loss landscape geometry and deep learning. With this in mind, we chose models to occupy a ‘sweet spot’ between interpretable toy models and practical LLMs.
>
> * Prior work \[1\] studied the link between geometry and development in a toy autoencoder with around 20 parameters. Because of the small size, they could analyze this model’s loss landscape and the development of its internal structure in precise detail, and the connection between geometry and development was clearly demonstrated.
> * In this paper, we studied models 2,500x to 150,000x larger trained on substantially more complex data (in-context linear regression, natural language). Our models are of comparable scale to those studied in other recent published works on the science of deep learning \[2,3\]. This work provides useful prior knowledge about behavioral/structural developments that we can use to validate that our geometric stage divisions are meaningful.
> * Analyzing a substantially larger model (e.g. on the order of BERT or a GPT) would offer a more direct demonstration of the practical implications of the links between geometry and development. However, we expect that in larger models the manifestation of this link will be more complicated than the clear developmental stages identified in the present work and may require the introduction of more refined techniques (see e.g. line 497 (was 480\) of the paper).
>
> Our paper demonstrates that there is a link between loss landscape geometry (as measured by the LLC) and neural network development in transformers at the scale of 3M parameters. At this scale, the link can be summarized by the clear developmental stages that we study. As the scale of models increases, we observe that the developmental stages become less clear, with our hypothesis being that larger models learn multiple things in parallel; for example the stages in the present work are more gradual than the sharp transitions in the smaller models studied in \[1\]. At some scale we expect that the link between loss landscape geometry and development is better described by a more refined notion of developmental stage, and the simple methodology in the present paper may no longer be applicable.
>
> Nonetheless we view the present paper as providing an essential foundation for the development of more sophisticated techniques that are applicable to larger models, since it validates the basic premise carefully in smaller models where the ground truth is easier to access.
>
> **LLC estimation away from local minima:** Reviewers PMP3 and E5MY also noted that a rigorous theoretical justification for estimating the LLC away from local minima is currently missing. The meaningfulness of LLC estimates in these cases is empirically supported by \[1\]. We agree that studying the LLC away from local minima would be valuable and represents an important future direction, though we consider it beyond our scope, and have explicitly acknowledged it as a limitation of the current work.
>
> **Thank you again.** We look forward to the remainder of the discussion.
>
> \[1\] Chen et al. (2023); \[2\] Olsson et al. (2022); \[3\] Raventós et al. (2023); as cited in manuscript.

---

### Author Response · Authors · 2024-11-18
**Guide to the revision**

Dear reviewers,

We are uploading a revised version of the manuscript with the following non-trivial changes, as noted in various places in the discussion:

1. Rewrote the abstract and introduction to reflect clarifying discussion in the top-level comment about the primary takeaways of the work.
2. Merge table 1 and figure 1 into a single figure with slightly more detailed caption (all subsequent table numbers reduced by 1).
3. Minor updates to improve the readability of section 2 on the two learning problems.
4. Rewrote section 3 “Methodology” (now spans sections 3 and 4, all subsequent section numbers in main text increased by 1).
   1. New section 3 introduces the LLC, LLC estimation, and prominently acknowledges its potential limitations.
   2. New section 4 explains the singular learning process and motivates our stage-identification methodology (why we use d(LLC)/dt \= 0 to mark stage boundaries).
5. Two new conceptual figures, figures 2 and 3, to illustrate the LLC and the singular learning process (all subsequent figure numbers in main text increased by 2).
6. Minor changes to sections 5 and 6 (was: sections 4 and 5), for example to clarify stage LM3 interpretation and other similar clarificatory changes.
7. Rewrote section 7 (was: section 6\) “Discussion and related work” (now spans sections 7 and 8).
   1. Section 7 “related work” discusses related work in more detail.
   2. Section 8 “discussion” now discusses the four main takeaways from the results in substantial detail.
8. Minor changes to appendix A to clarify limitations of our LLC estimation methodology.
9. Minor changes to appendix C to clarify the relationship between degeneracy and layer normalization (and other topics).

Note that at this stage all changes affect only the presentation of the results, and none of the experiments or results themselves have been updated or altered in any way. We hope that the refreshed presentation in the introduction, methodology sections, and discussion in particular help you more clearly appreciate our work.

Please let us know if you have any questions or concerns about the rewrite.

---

### Meta-Review · Area_Chair_JpQb · 2024-12-20

**Metareview:**

This paper explores how transformers evolve during training by analyzing the geometry of the loss landscape using the Local Learning Coefficient (LLC). It identifies distinct developmental stages marked by shifts in LLC, coinciding with changes in model behavior and internal structure, suggesting a link between loss landscape geometry and model development. Strengths include a novel application of LLC to identify developmental stages in transformers and validation across two different tasks, offering new insights into transformer training dynamics. However, the reliance on the relatively unvetted LLC metric and the limited scope of small, attention-only transformers raise concerns about the generalizability and robustness of these findings, especially to larger, more complex models used in practice, ultimately warranting rejection to encourage more rigorous validation and broader applicability.

**Additional Comments On Reviewer Discussion:**

Reviewers questioned the generalizability of findings from small transformers to larger models, the theoretical grounding of using LLC away from local minima, the novelty of insights beyond existing work, and various experimental details. Authors responded that their core contribution is demonstrating a link between loss landscape geometry and development, with small models chosen for interpretability. They defended the use of LLC, citing empirical validity and its basis in singular learning theory, and asserted that their work extends previous findings by revealing a progression of strategies in early training stages and providing a complementary perspective through LLC, which reveals changes hidden from standard loss metrics. They also provided clarifications on experimental procedures and highlighted other contributions, such as new insights into transformer development and a method for developmental interpretability.

While some reviewers found the rebuttal compelling, in the end the reviewers were split on their recommendations. Ultimately, given the number of potentially outstanding concerns, the community would be better served by a future version of this manuscript that incorporates more of the reviewers' suggestions.

---

### Decision · Program_Chairs · 2025-01-22

Reject